# Safety and Anatomical Accuracy of Dry Needling of the Quadratus Femoris Muscle: A Cadaveric Study

**DOI:** 10.3390/healthcare13151828

**Published:** 2025-07-26

**Authors:** Marta Sánchez-Montoya, Jaime Almazán-Polo, Néstor Vallecillo Hernández, Charles Cotteret, Fabien Guerineau, Domingo de Guzman Monreal-Redondo, Ángel González-de-la-Flor

**Affiliations:** 1Department of Medicine, Faculty of Medicine, Health and Sports, European University of Madrid, Villaviciosa de Odón, 28670 Madrid, Spain; marta.sanchez@universidadeuropea.es (M.S.-M.); nestor.vallecillo@universidadeuropea.es (N.V.H.); dguzman.monreal@universidadeuropea.es (D.d.G.M.-R.); 2Faculty of Nursing, Physiotherapy and Podiatry, Universidad Complutense de Madrid, 28040 Madrid, Spain; 3Department of Physiotherapy, Faculty of Medicine, Health and Sports, European University of Madrid, Villaviciosa de Odón, 28670 Madrid, Spain; charles.cotteret@universidadeuropea.es (C.C.); fabien.guerineau@universidadeuropea.es (F.G.); angel.gonzalez@universidadeuropea.es (Á.G.-d.-l.-F.)

**Keywords:** dry needling, quadratus femoris muscle, cadaveric study, ischiofemoral space, sciatic nerve, reliability, ultrasound imaging, anatomic dissection

## Abstract

**Introduction:** Deep dry needling (DDN) is commonly applied in physiotherapy to treat musculoskeletal pain. The quadratus femoris (QF) muscle, located in the ischiofemoral space (IFS), represents a clinically relevant yet anatomically complex target. However, limited evidence exists on the safety, accuracy, and reliability of non-ultrasound-guided DDN in this region. **Aims:** To assess the safety and accuracy of a standardized, non-ultrasound-guided DDN approach to the QF muscle, and to evaluate the intra- and inter-rater reliability of key procedural outcomes. Additionally, to determine the agreement between ultrasound imaging and anatomical dissection as validation methods for needle placement. **Methods:** An experimental cross-sectional study was conducted on five fresh cadavers (n = 24 approaches) by two physiotherapists with different DN experience. A standardized dry needling protocol was executed without ultrasound guidance, and anatomical and procedural variables were documented. Reliability (intra/inter-rater) was assessed for needle size, sciatic nerve (SN) puncture, IFS targeting, and overall success. In a subset, needle placement was validated through ultrasound and subsequent dissection. **Results:** The IFS was reached in 70.8% of procedures, and the SN was punctured in 16.7%. Inter-rater reliability for needle size was poor (κ = 0.04). Agreement between ultrasound and dissection was excellent for the ischiofemoral location and success (100%) and moderate for non SN puncture (90%; κ = 0.62). **Conclusions:** The standardized protocol demonstrated moderate accuracy and revealed a relevant clinical risk when targeting the quadratus femoris muscle. While inter-rater reliability was limited, agreement between ultrasound and dissection methods was high, supporting their complementary use for validating needle placement in anatomically complex procedures.

## 1. Introduction

The quadratus femoris (QF) muscle has gained increasing attention as a potential source of deep gluteal pain, particularly in conditions involving narrowing of the ischiofemoral space (IFS) or as part of deep gluteal syndrome, a broader classification that encompasses various causes of non-neurogenic sciatic nerve entrapment in the subgluteal space [1]. Clinical syndromes such as ischiofemoral impingement, often characterized by buttock pain exacerbated by hip flexion and referring posterior thigh discomfort, have been associated with mechanical irritation of the sciatic nerve due to QF involvement [2].

Anatomically, the QF is innervated by the nerve to quadratus femoris, which also supplies the posterior and posterolateral regions of the hip capsule, suggesting a potential role in referred hip pain of capsular origin. The muscle originates from the lateral border of the ischial tuberosity and inserts onto the intertrochanteric crest of the femur, functioning primarily as a hip external rotator and assisting in adduction [3,4]. Within this anatomical and clinical context, deep dry needling has emerged as a physiotherapeutic intervention with potential value in modulating neuromuscular dysfunction in deep gluteal structures. However, the anatomical complexity and the close relationship of the QF to critical neurovascular elements, such as the sciatic nerve (SN), present significant technical challenges [5]. In particular, when performed without ultrasound guidance, the accuracy and safety of deep dry needling in this region remain uncertain, highlighting the need for procedural standardization and reliability analysis in anatomically sensitive areas [5].

Dry needling typically involves the use of solid stainless-steel monofilament needles with a sharp tip, selected according to the depth and location of the target tissue [6]. Among physiotherapeutic interventions, deep dry needling is increasingly employed to target myofascial trigger points or structurally relevant deep tissues, such as the quadratus femoris, especially when managing conditions related to deep gluteal pain. Although ultrasound guidance can enhance accuracy and safety, many procedures are still performed blindly in clinical settings, raising important concerns regarding the reliability of targeting and the potential risk of injury to nearby structures [7].

From a methodological perspective, two primary tools are available to validate needle placement in deep dry needling procedures: musculoskeletal ultrasound and anatomical dissection [8]. Ultrasound imaging enables real-time visualization of the needle tip and its relationship to adjacent structures, while dissection offers a direct assessment of the final needle positioning in cadaveric specimens [9]. Despite their utility, no prior studies have compared the degree of agreement or reliability between these two methods when used to assess the success of invasive procedures targeting deep anatomical structures. Although a few studies have explored the accuracy and safety of dry needling in deep or anatomically complex regions—such as the popliteus, pronator quadratus, medial pterygoid, tibialis posterior, and the distal tendon of the biceps brachii, among others—evidence remains scarce regarding the inter-rater reliability and agreement on clinical success between different practitioners [10,11,12,13,14].

The present study aims to evaluate the safety and accuracy of a standardized, non-ultrasound-guided deep dry needling approach to the quadratus femoris muscle in a cadaveric model. Secondary objectives include assessing the intra- and inter-rater reliability of key anatomical and procedural variables across different needle insertions, as well as determining the level of agreement between ultrasound imaging and anatomical dissection in validating needle placement. Reliability was evaluated between two physiotherapists with differing levels of clinical experience in deep dry needling, both of whom had extensive teaching experience in musculoskeletal anatomy within undergraduate physiotherapy education. We hypothesize that, despite adherence to a closed step-by-step protocol, the absence of ultrasound guidance may limit the accuracy of the technique, and that reliability metrics will vary depending on examiner experience and the method of validation.

## 2. Methods

### 2.1. Study Design

An experimental, cross-sectional, and analytical study was conducted using fresh human cadaveric specimens to evaluate the reliability and accuracy of a dry needling protocol targeting the quadratus femoris muscle. This study adhered to the methodological quality criteria outlined by the QUACS (Quality Appraisal for Cadaveric Studies) scale, including a clear statement of objectives, detailed methodological reporting, description of specimen condition, and documentation of researcher qualifications and inter-observer consistency. The study protocol received ethical approval from both the Research Commission of Universidad Europea de Madrid (approval code: 2025-30) and the Clinical Research Ethics Committee of Hospital Clínico San Carlos (approval code: C.I. 25/157-E).

### 2.2. Cadaveric Specimens and Sample Characteristics

A total of five fresh cryopreserved human cadavers were used, including both lower limbs (right and left), yielding a total of 24 independent procedures. Fresh-frozen specimens were selected due to their high anatomical and biomechanical fidelity, which allows for detailed dissection and precise visual assessment of the needle trajectory and final positioning, offering conditions that closely resemble those of living tissues [6,14,15]. The final sample selection was guided by previous literature on cadaveric research involving similar anatomical targets, as well as by established recommendations from clinical guidelines for cadaveric experimental design [16,17,18]. In accordance with the QUACS criteria, efforts were made to ensure sex parity; however, due to operational limitations, only one female specimen was included. Specimens were screened to exclude any pathological or traumatic alterations that could potentially affect the quadratus femoris muscle or adjacent anatomical structures.

All cadavers were obtained through a regulated body donation program and were designated for research and educational purposes at Universidad Europea de Madrid. Procurement and handling procedures adhered to current ethical and legal standards. Prior to the intervention, specimens were thawed at room temperature for 72 h to ensure optimal tissue consistency for needling and anatomical evaluation.

### 2.3. DDN Procedure and Standardization Protocol

All needling procedures were performed by two licensed physiotherapists with differing levels of clinical experience in DDN and certified postgraduate training in invasive physiotherapy techniques. Experience levels were defined as intermediate (Ph.1, 5 years) and high (Ph.2, over 10 years). Both physiotherapists also held extensive teaching experience in human anatomy, serving as faculty in the Department of Physiotherapy at Universidad Europea de Madrid.

Fifteen minutes before the intervention, physiotherapists received a written copy of the needling protocol to ensure consistency, which included detailed step-by-step instructions on needle placement. The document provided guidance on anatomical landmarks, measurement techniques for determining the entry point, and recommended needle inclination angles relative to cadaver positioning. Additional support was provided through visual aids, including multi-planar anatomical illustrations of the target region and diagrams of surrounding critical structures to avoid during insertion. To standardize protocol acquisition, a member of the research team presented the protocol aloud, clarified any questions raised by the clinicians, and highlighted key procedural steps. Visual support included on-screen images of the protocol displayed in the dissection room, allowing real-time review of anatomical references and technical instructions. Physiotherapists were also encouraged to ask questions during the needling procedures if clarification was required (Appendix A). Only two aspects were left to the clinician’s discretion: (1) the selection of the needle size and thickness based on cadaveric dimensions, and (2) the depth of needle insertion, which was evaluated exclusively for Ph.2.

Each interventionist performed their procedures individually in the dissection laboratory, without observation or interference from the other rater. The cadavers were systematically arranged and labeled from 1 to 5 on separate dissection tables. All specimens were placed in the prone position, with the feet hanging freely beyond the edge of the table to prevent passive plantar flexion. To maintain a neutral ischiofemoral space and minimize post-mortem external rotation, a positioning block was placed under each pelvis to stabilize the hip in neutral rotation. Ph.1 completed a full round of bilateral dry needling insertions across all cadavers, performing a total of 10 procedures targeting the quadratus femoris muscle. Ph.2 conducted two rounds: the first involved bilateral insertions on all cadavers (10 procedures), and the second included four additional insertions on cadavers 1 (female) and 3 (male) for intra-rater reliability analysis. In accordance with the QUACS guidelines for cadaveric research, an external observer familiar with the protocol was present during all procedures to ensure that each interventionist adhered strictly to all steps of the standardized protocol (Figure 1).

### 2.4. Ultrasonographic Assessment of DDN Procedure

Following each insertion, a physiotherapist with over 10 years of clinical experience in musculoskeletal ultrasound, who was blinded to the physiotherapists’ needle insertion, performed a real-time sonographic evaluation using a high-resolution ultrasound system (Logiq S7 Expert, GE Healthcare, Chicago, IL, USA) to assess the success of the DDN procedure. A single broad-spectrum linear matrix array probe (ML6-15, 50 mm field of view) with a frequency range of 5–15 MHz was used for all measurements.

The following outcome variables were assessed: (1) needle tip position within the ischiofemoral space (success/failure); (2) needle tip position within the quadratus femoris muscle (success/failure); (3) vascular bundle puncture (success/failure); (4) sciatic nerve puncture (yes/no); and (5) needle size selected (0.30 × 30 mm; 0.30 × 40 mm; 0.30 × 50 mm; 0.30 × 60 mm). The first three were classified as dichotomous variables (success or failure), the fourth as a binary variable (yes/no), and the fifth as a multicategorical qualitative variable. Needle placement within the ischiofemoral space (IFS) was operationally defined as the location of the needle tip between the lateral margin of the ischial tuberosity and the posterior edge of the greater trochanter, corresponding to the target region specified in the protocol. Insertions located outside this anatomical corridor, such as overlying bony structures or positioned proximally or distally to the defined boundaries, were considered failures. For the purposes of this study, global procedural success (success = yes) was defined as those cases in which the needle tip reached the QF muscle, was located within IFS (specifically in the lateral region targeted by the needling approach, as detailed in the standardized protocol; Appendix A), and did not puncture the SN. Any deviation from these three criteria resulted in the classification of the procedure as unsuccessful. This operational definition was intended to reflect both anatomical accuracy and procedural safety.

Quantitative ultrasound measurements were obtained using the system’s integrated electronic calipers and were expressed in centimeters, including: (6) distance from the needle tip to the sciatic nerve; (7) gluteal skin thickness; and (8) total depth of needle insertion (Figure 2). After completion of the sonographic evaluations, cadavers were preserved for subsequent anatomical dissection to confirm final needle placement. For the assessment of intra-rater reliability, Ph.2 repeated the procedures on cadavers 1 and 3, which corresponded to female and male specimens, respectively. This selection aimed to ensure sex parity and optimize specimen quality for evaluating procedural consistency. The final four interventions performed by Ph.2 on these cadavers were used for anatomical validation, with the needles left in situ to facilitate direct verification during dissection. To determine the needle position, several ultrasound functionalities were employed, including the needle vision system, the virtual convex option to widen the insonation field and enhance lateral echo visualization, and adjustments to the insonation angle to optimize needle visibility. Additionally, power Doppler was activated while vibrating the needle to amplify signal detection and accurately localize its position (Appendix A).

### 2.5. Disecction Assessment of DDN Procedure

Upon completion of the sonographic assessments, the cadavers were preserved for subsequent anatomical dissection to confirm final needle placement. For this purpose, the needles were maintained in situ in the last ten procedures performed by Ph.2 on cadavers 1 and 3 for intra-rater assessment, as well as cadavers 2, 4, and 5 during the final approaches, enabling precise anatomical validation of needle trajectory and target localization. Anatomical dissections were then conducted by a team of two anatomists in collaboration with a third senior faculty member with extensive experience in dissection techniques. All three were academic staff in the Department of Anatomy at the School of Medicine, European University of Madrid, each with over five years of teaching experience.

To preserve the exact position of the needle during dissection, the exposed needle shaft was stabilized using cyanoacrylate adhesive, preventing displacement during specimen handling and tissue separation. The needle was left in situ throughout the dissection procedure to evaluate its trajectory and identify the anatomical structures traversed prior to reaching the QF muscle. Dissection involved progressive removal of skin, fascia, and superficial musculature from the deep gluteal region and ischiofemoral space. Following subcutaneous tissue removal, the gluteus maximus was identified and partially removed, along with superficial fibers of the gluteus medius, to access the deep gluteal region. The peritrochanteric muscles were dissected and cleaned of adipose tissue until the underlying anatomical structures became visible [4,19]. The SN was carefully dissected starting from the piriformis level and used as a reference to locate the needle’s path. Special attention was paid to preserving neurovascular structures, particularly the inferior gluteal nerve and artery, and the SN (including the tibial and common fibular branches, when present). Subsequently, the quadratus femoris muscle and surrounding structures were identified and color-coded for documentation (blue: quadratus femoris; orange: inferior gemellus; white: obturator internus; black: superior gemellus; red: piriformis; yellow: sciatic nerve; black arrow: dry needling needle) (Figure 2).

### 2.6. Statistics

The statistical analysis was performed using IBM SPSS Statistics (v.30.0), with the significance level set at α = 0.05. Descriptive statistics were applied to all demographic, anthropometric, and procedural variables, including age, height, weight, estimated body mass index (BMI), and hip perimeter, all of which were obtained from the five cadavers included in the study (n = 5). Continuous variables were tested for normality using the Shapiro–Wilk test, supported by visual inspection, skewness, and kurtosis. Parametric data were summarized as mean ± standard deviation (SD), 95% confidence interval (CI), and minimum–maximum range, while non-parametric data were reported as median, interquartile range (IQR), and minimum–maximum range. Categorical variables were expressed as frequencies and percentages (%).

Each cadaveric approach (Event, n = 24) was recorded individually, specifying the physiotherapist performing the procedure (Ph.1, 3 years of experience; Ph.2, 10 years of experience), the side of the body on which the approach was performed (side, right or left), the cadaver (Cadaver ID, coded from 1 to 5, and a or b based on the right or left side, respectively), laterality (side, right or left), and sex (male or female). Procedural and anatomical variables included needle size selection, needle depth (ND), subcutaneous fat thickness (Sub.Fat.Th), gluteus maximus thickness (G.Max.Th), and distance to the sciatic nerve (SN.Dis.), which were considered continuous quantitative variables. Of note, ND was only measured and reported by Ph.2. Outcome variables included ischiofemoral space location (IFS), sciatic nerve puncture (SN.Punct.), and global procedural success (Success), all classified as nominal dichotomous variables and recorded as “yes” or “no” based on predefined binary criteria. Needle size was treated as a nominal multicategorical variable with four predefined categories (0.30 × 30 mm, 0.30 × 40 mm, 0.30 × 50 mm, and 0.30 × 60 mm).

Intra- and inter-rater reliability were assessed for dichotomous variables (IFS, SN.Punct, Success) and the multicategorical variable Needle Size. For all variables, the observed percentage of agreement was calculated as the number of exact matches divided by the total number of observations and multiplied by 100 (% agreement = [matches/total] × 100). For Needle Size, the most frequently selected value (mode) and its relative frequency (mode frequency = [mode count/total] × 100) were also computed. Cohen’s kappa coefficient (κ) was calculated to assess agreement beyond chance, and 95% CIs were derived using the standard error based on the binomial approximation (CI = κ ± 1.96 × SE). Kappa values were interpreted according to the Landis and Koch scale, which classifies agreement as follows: κ < 0.00 = no agreement (worse than chance), 0.00–0.20 = poor agreement, 0.21–0.40 = fair agreement, 0.41–0.60 = moderate agreement, and 0.61–0.80 = high agreement [20]. When one rater showed no variability in responses, κ was computed as 0.000 and considered not interpretable (N/I) due to the absence of statistical variance. All reliability values were reported together with the number of observations (N), the 95% confidence interval (CI), and the *p*-value, when interpretable.

Finally, agreement between ultrasound imaging and cadaveric dissection was assessed for the three dichotomous outcome variables (IFS, SN.Punct., and Success). For each comparison, the observed agreement, Cohen’s kappa coefficient with its 95% confidence interval, and the Valence Index were calculated, using the formula: Valence Index = |P_US_ − P_DIS_|, where P_US_ and P_DIS_ represent the proportion of positive outcomes identified by ultrasound and dissection, respectively. This analysis was conducted exclusively on the final approach performed by Physiotherapist 2 (clinical experience of 10 years), for whom both ultrasound and anatomical confirmation were available, yielding a total of 10 procedures (n = 10).

## 3. Results

### 3.1. Decriptive Cadaver Characteristics

A total of five fresh-frozen human cadavers (n = 5) were included in the study. The mean age was 79.6 ± 8.0 years, and four cadavers were male (80%). The average height was 1.63 ± 0.11 m, and the mean weight was 60.6 ± 13.0 kg. The estimated BMI was 22.7 ± 2.8 kg/m^2^, ranging from 19.3 to 26.8. Hip perimeter measurements averaged 92.0 ± 4.2 cm (Table 1).

### 3.2. Descriptive Analysis of Procedural and Anatomical Variables

A total of 24 deep dry needling procedures were performed without ultrasound guidance (Table 2). Each procedure was documented by cadaver ID (1–5), side (right/left), Ph.1 or Ph.2, and outcome. Ph.1 performed 10 procedures, while Ph.2 performed 14 procedures. Specifically, Ph.2 conducted all approaches on cadavers 1 and 3 (both right and left sides), which were in better preservation condition and represented both sexes (one male, one female) to ensure anatomical diversity and gender parity. Of the 24 procedures, Ph.1 performed 10 (41.7%), while Ph.2 performed 14 (58.3%). Despite efforts to ensure gender parity, only one female cadaver was available, resulting in a male-to-female ratio of 3:1 and 75% of procedures were conducted on male specimens. A summary of the 24 invasive procedures is shown in Table 3. The most frequently selected needle size was 0.30 × 60 mm (n = 12; 50%), followed by 0.30 × 40 mm (n = 8; 33.3%). The remaining selections included 0.30 × 30 mm (n = 2; 8.3%) and 0.30 × 50 mm (n = 2; 8.3%). Regarding stratification by examiner, Ph.1 selected 0.30 × 40 mm in 8 out of 10 procedures (80%) and 0.30 × 50 mm in 2 procedures (20%), whereas Ph.2 selected 0.30 × 60 mm in 12 out of 14 procedures (85.7%) and 0.30 × 30 mm in 2 procedures (14.3%). Thus, each examiner consistently selected only two of the four available needle sizes, with no overlap between them. In terms of binary outcomes, 17 out of 24 procedures (70.8%) were rated as globally successful (Success = yes)—3 by Ph.1 (23.1%) and 10 by Ph.2 (76.9%). Ischiofemoral space location (IFS = yes) was achieved in 22 of 24 cases (91.7%), including 8 of 10 procedures by Ph.1 (36.4%) and 14 of 14 by Ph.2 (63.6%). Sciatic nerve puncture (SN.Punct. = yes) occurred in two procedures (8.3%), both performed by Ph.2 (100%). Regarding anatomical measurements, the Sub.Fat.Th. had a mean of 0.86 ± 0.22 cm (range: 0.43–1.22), the G.Max.Th. was 1.24 ± 0.33 cm (range: 0.79–2.06), and the SN.Dis. was 1.29 ± 0.96 cm (range: 0.00–3.45). Stratified by examiner, Ph.2 reported slightly higher average values in all three variables: Sub.Fat.Th. = 0.90 ± 0.20 cm, G.Max.Th. = 1.27 ± 0.43 cm, and SN.Dis. = 0.95 ± 0.93 cm.

### 3.3. Intra and Inter-Rater Reliability

Agreement analyses were performed for dichotomous and multicategorical variables (Table 4). Intra-rater reliability (Ph.1) showed full agreement for both IFS and SN puncture (100%), and moderate agreement for global success (50%; κ = 0.194; 95% CI: −0.306 to 0.490). In all three cases, the number of observations was n = 4, and no *p*-values were computed due to the absence of variability or low frequency of positive findings, leading to κ = 0.000 (not interpretable). For needle size selection, intra-rater agreement was 50%, with a mode frequency of 50% and a κ value of 0.000, not interpretable due to the absence of variability (N = 4; *p* = 1.000). Inter-rater reliability (Ph.1 vs. Ph.2) revealed no exact matches for needle size selection (0% agreement; κ = 0.000, not interpretable), with Ph.1 predominantly selecting 0.30 × 40 mm and Ph.2 preferring 0.30 × 60 mm. Despite a mode frequency of 60%, no agreement was observed, and κ = 0.000 was again deemed not interpretable (N = 10; *p* = 1.000). For the dichotomous outcomes, IFS and SN puncture each showed high observed agreement (80%), but κ was 0.000 and not interpretable due to a lack of variability in one rater’s responses (both N = 10; *p* = 1.000). Specifically, in the case of SN puncture, the 80% agreement corresponded to cases in which both raters reported no puncture of the sciatic nerve. For global success, inter-rater agreement was 50%, with κ = 0.194 and a 95% CI of –0.306 to 0.490 (N = 10; *p* = 0.301). In cases where κ was equal to 0.000 and not interpretable due to insufficient variability or complete disagreement, the corresponding *p*-value was considered to be 1.000 under the null hypothesis of chance agreement. These cases are indicated in Table 4 using the symbols † (N/I, not interpretable) and ‡ (*p*-value assumed as 1.000).

### 3.4. Agreement Between Ultrasound Imaging and Cadaveric Dissection

Agreement analysis between ultrasound imaging and cadaveric dissection was conducted on the last 10 procedures performed by Ph.2 (more than 10 years of experience), in which both confirmation methods were available (Table 5). Complete concordance was observed for IFS, with 100% agreement and a non-interpretable kappa value due to lack of variability (κ = 0.000; *p* = —; Valence Index = 1.000). For sciatic SN.Punct., 90% agreement was reached, corresponding to 9 out of 10 cases in which both methods confirmed the absence of nerve puncture. Cohen’s kappa indicated high inter-method agreement (κ = 0.615; 95% CI: 0.035–1.276; *p* = 0.035; Valence Index = 0.800). Regarding global procedural success, ultrasound and dissection showed full agreement (100%), supported by a perfect kappa score (κ = 1.00; 95% CI: 1.00–1.000; *p* = 0.002; Valence Index = 0.200. Significant inter-method agreement (*p* < 0.05) was found for both SN.Punct. and global success, while the agreement for IFS was perfect but not statistically interpretable due to the absence of variability. These findings support the validity of ultrasound as a reliable alternative to anatomical dissection in assessing these procedural outcomes.

## 4. Discussion

### 4.1. Overview of Findings

This study aimed to evaluate the accuracy and safety of a DDN approach to the QF muscle through a structured and standardized step-by-step protocol in a cadaveric model. The IFS was successfully reached in 70.8% of procedures, while SN puncture occurred in 16.7% of cases. Although these findings reflect a potentially acceptable anatomical profile in terms of safety, the puncture rate may still be considered clinically relevant. Moreover, the fact that nearly one-third of insertions failed to reach the intended target highlights important limitations in procedural accuracy, even under controlled conditions. These results underscore the need for cautious interpretation when considering the implementation of blind techniques in clinical practice.

To the best of the authors’ knowledge, this is the first study to systematically evaluate the reliability of DDN procedures targeting the QF muscle in a cadaveric model, while simultaneously validating procedural outcomes through dual assessment using both ultrasound imaging and anatomical dissection. Although previous studies have employed both modalities to verify needle positioning [11,21], none have specifically aimed to assess the inter-method agreement between ultrasound and dissection alongside inter-examiner reliability metrics. By combining reproducibility metrics across evaluators with differing levels of clinical experience and inter-method agreement between imaging and dissection, the present findings provide novel insights into the technical feasibility, anatomical risks, and standardization challenges associated with invasive procedures in high-risk anatomical zones. The absence of prior data for direct comparison underscores the originality of this approach and highlights the need for further research into adjunct strategies such as ultrasound guidance, protocol refinement, and targeted training programs to improve procedural accuracy, safety, and reproducibility in both clinical and educational contexts.

In recent years, several cadaveric studies have described invasive approaches to anatomically challenging regions, primarily focusing on needle placement accuracy in relation to target tissues and critical neurovascular structures. These investigations have included muscles such as the pronator quadratus, pronator teres, supinator, lateral pterygoid, popliteus, piriformis, and tibialis posterior, among others [6,10,13,14,15,21]. While this methodological approach constitutes a strength of the present study, offering a novel contribution to the literature, it simultaneously limits the possibility of direct comparisons with prior research. Nevertheless, these findings align with recent evidence supporting the safety of non-guided DDN when standardized protocols are followed, although ultrasound may be advisable in high-risk regions due to methodological limitations in previous studies [22].

### 4.2. Safety, Accuracy and Reliability of the Procedure

Despite using a closed and replicable protocol, a considerable percentage of procedures (29.2%) failed to reach the IFS space, underscoring the inherent difficulty of accessing deep anatomical targets without image guidance [1]. Moreover SN.Punct. occurred in 16.7% of the procedures, which must be considered a clinically relevant risk, especially in real-life scenarios involving patient discomfort, movement, or anatomical variability [5]. These findings highlight that although the technique may present an anatomically feasible approach when guided by external landmarks and consistent methodology, its accuracy and safety cannot be guaranteed without imaging support. The close proximity of critical neurovascular structures demands a high level of anatomical expertise and reinforces the importance of controlled training environments. In this sense, the observed intra and inter-rater variability, especially regarding needle size selection, reflects the potential for inconsistency even among experienced physiotherapists. Similar concerns have been previously raised regarding the safety of deep needling in anatomically complex regions, such as the piriformis muscle, for which ultrasound guidance has been explicitly recommended to avoid unintended neurovascular injury [23].

Several factors may have contributed to the observed inaccuracies. Although fresh-frozen cadavers offer high fidelity for simulating in vivo conditions, tissue stiffness and dehydration can alter mechanical resistance, potentially influencing needle trajectory during insertion. In particular, prone storage positions appeared to displace muscle masses, such as the gluteus maximus, flattening the surface topography. According to the external observer supervising protocol adherence, this distortion occasionally impaired the identification of key bony landmarks, including the lateral edge of the ischial tuberosity and the posterior margin of the greater trochanter. Despite the physiotherapists’ solid anatomical background and clinical palpation expertise, such alterations may have led to subtle misjudgments in entry point localization and needle orientation.

Previous cadaveric investigations have reported high accuracy in targeting deep or anatomically complex muscles—such as the pronator quadratus, tibialis posterior, supinator, or popliteus—using ultrasound guidance or post-procedure validation [10,13,14]. However, these studies typically relied on single-method validation and did not incorporate inter-rater reliability or systematic procedural replication. In contrast, the present study employed a more comprehensive methodological framework that included adherence to QUACS guidelines, the use of a detailed step-by-step needling protocol, and dual assessment via ultrasound and dissection [16]. Although needle size selection and depth of insertion were left to clinician discretion to simulate real-world variability, the overall protocol structure enhanced standardization and reproducibility.

Regarding reliability assessment, intra- and inter-rater analyses revealed substantial differences between evaluators. Intra-rater agreement for dichotomous variables (IFS, SN.Punct., and success) was highest for Ph.2, who had greater clinical experience in DDN (10 years), achieving 100% agreement for both IFS and SN.Punct. However, overall procedural success was only 50%. This apparent discrepancy may be explained by the fact that, although all procedures reached the IFS space, in two out of four cases the needle failed to penetrate the quadratus femoris muscle, remaining within its superficial fascia. Such misplacements may be attributed to the increased stiffness of cadaveric tissues, which can alter the clinician’s perception of tissue transitions during insertion. In non-ultrasound-guided procedures, needling accuracy often relies on tactile feedback and variations in resistance to identify anatomical planes, but this sensory cue may be distorted in preserved specimens, compromising placement reliability.

For the multicategorical variable of needle size selection, intra-rater reproducibility was limited (50% agreement; mode frequency = 50%), as Ph.2 selected a different needle in the second round. This adjustment may reflect clinical reasoning based on perceived tissue depth or insertion resistance. In contrast, inter-rater agreement was 0%, with Ph.1 consistently selecting the intermediate length (0.30 × 40 mm), while Ph.2 opted for the longest needle (0.30 × 60 mm). These differences suggest that material selection may be influenced by each clinician’s spatial judgment and professional experience, as well as subjective risk perception. Beyond the reliability analysis, a noteworthy observation was made during the review of individual cases, as Ph.2 selected a shorter 0.30 × 30 mm needle for Cadaver 4 despite the anatomical depth of the target muscle. This isolated choice may be explained by the condition of the specimen (height: 1.62 m; weight: 55 kg), which was notably the smallest among all cadavers. Due to severe muscular atrophy and pronounced tissue flattening, bony landmarks were more prominent and superficially palpable than in typical in vivo conditions. This altered perception could have led the clinician to underestimate the required depth and opt for a shorter needle. Such considerations highlight the methodological limitations of extrapolating needling choices in preserved specimens to real-world clinical practice, especially when dealing with older cadavers exhibiting substantial anatomical degradation. Importantly, the lack of prior consensus or training on needle size selection was intentionally incorporated into the study design to explore individual decision-making under realistic clinical conditions. This approach likely contributed to the poor inter-rater reliability observed (κ = 0.04), underscoring the need for standardized selection criteria in future applications. With regard to the dichotomous outcomes, the agreement observed was high (80%) for both IFS location and SN.Punct., although κ was 0.000 in both cases due to the absence of variability in one rater’s assessments. Notably, the 80% agreement for SN.Punct. was entirely attributable to instances where both evaluators reported no SN puncture, thereby supporting the overall safety of the approach. Additionally, none of the reliability comparisons yielded statistically significant differences (all *p* ≥ 0.301), further confirming the consistency of the observed patterns across raters.

### 4.3. Agreement Between Ultrasound Imaging and Anatomical Dissection

The comparative analysis between ultrasound imaging and cadaveric dissection revealed high levels of agreement for all evaluated variables. In the case of IFS, perfect concordance was observed (100% agreement), although Cohen’s kappa was not interpretable due to the complete absence of variability across both methods (κ = 0.000). The Valence Index reached its maximum value (1.000), reflecting the asymmetry produced by a homogeneous outcome distribution. While such results confirm procedural consistency, they also highlight the limitations of kappa when outcome prevalence is extreme. In line with this, the *p*-value associated with the kappa statistic was 1.000, confirming the lack of statistical interpretability in this case. In a similar cadaveric study on the shoulder, substantial to almost perfect inter-examiner agreement between physiotherapists, radiologists and dissection was demonstrated when using musculoskeletal ultrasound to identify pathology in rotator cuff structures, supporting the validity of sonographic assessment in anatomically complex regions [24]. Additionally, recent in vivo findings have shown that palpation-based dry needling techniques targeting the supraspinatus and infraspinatus muscles can achieve high anatomical accuracy when verified with ultrasound, further supporting the use of sonographic imaging as a valuable tool for procedural validation in deep musculature [25].

In the case of SN.Punct., 90% agreement was observed, and Cohen’s kappa indicated substantial intermethod reliability (κ = 0.615; 95% CI: 0.035–1.276), despite the relatively small sample size. Importantly, 9 out of 10 cases showed concordance in reporting an absence of nerve puncture, reinforcing the safety profile of the technique under anatomically controlled conditions. Nevertheless, the Valence Index (0.800) revealed a notable imbalance in outcome distribution, which could partially inflate the agreement statistics. The kappa result was statistically significant (*p* = 0.035), supporting the existence of true agreement beyond chance. As for global procedural success, ultrasound and dissection also demonstrated complete agreement (100%), reflected by a perfect kappa score (κ = 1.000; 95% CI: 1.000–1.000). Moreover, the Valence Index (0.200) remained low, suggesting a more balanced distribution of outcomes and lending further credibility to the observed agreement. This finding was further supported by a statistically significant *p*-value (*p* = 0.002), indicating perfect intermethod concordance.

These results must also be interpreted in light of specific technical challenges inherent to ultrasound guidance and cadaveric dissection procedures, which may help explain the few observed discrepancies. Despite the high degree of reliability observed, some discrepancies between ultrasound and dissection findings may be explained by technical and procedural factors. In several cases, accurate needle visualization by ultrasound required clear identification of the needle tip, which could be challenging when needle angulation deviated from the standardized insertion path. Under such circumstances, adjustments in probe tilt or insonation angle were needed to regain tip visibility, potentially introducing interpretation bias. Furthermore, the needle trajectory for the QF muscle was based on an out-of-plane sonographic approach, where tip recognition is critical but susceptible to confusion with the needle shaft, especially under suboptimal imaging conditions. Such limitations are well documented in ultrasound-guided procedures, where out-of-plane techniques in particular are known to increase the risk of misinterpretation unless specific imaging strategies are employed [25,26]. On the other hand, although anatomical dissection is considered the reference method, it also relies on maintaining needle fixation. Despite all precautions, minor needle displacement during handling or exposure could not be fully ruled out, representing an additional source of potential mismatch with real-time ultrasound confirmation.

### 4.4. Clinical Relevance and Rationale for Protocol Development

Despite differences in needling experience, both physiotherapists possessed high anatomical expertise, as faculty members actively engaged in undergraduate anatomy education and cadaveric dissection. This expertise, combined with the implementation of a closed and standardized protocol, provided an optimal setting to assess procedural reliability. Nevertheless, a 29.2% failure rate in accessing the target region was observed, highlighting the technical challenges of performing deep dry needling in anatomically complex areas. These findings suggest that anatomical knowledge alone may be insufficient to ensure consistent accuracy in blind procedures, particularly in the absence of complementary guidance systems. This concern is further reinforced by the current lack of standardization in dry needling protocols and the inconsistent documentation of adverse events in clinical trials, which limit the generalizability of safety and efficacy outcomes reported in the literature [27].

In support of this, Valera-Calero et al. [28] reported that 85.5% of 422 physiotherapists performing deep dry needling (DDN) did not routinely use ultrasound guidance, especially among those with less than two years of experience. In contrast, clinicians with more than ten years of experience were more likely to apply DDN in 80–100% of their patients, but only 14.5% reported using ultrasound regularly. These data emphasize the clinical relevance of investigating the safety and accuracy of blind procedures, particularly in anatomically risky regions such as the deep gluteal space. In the present study, the more experienced physiotherapist (Ph.2) demonstrated greater reliability and procedural accuracy, suggesting that clinical expertise may enhance safety and precision when ultrasound is not available. This further supports the implementation of structured, validated protocols to guide practice and reduce variability across different levels of clinical experience. Moreover, most invasive approaches in routine clinical settings continue to be performed without ultrasound guidance, likely due to time constraints and the greater technical demands associated with imaging, which require advanced sonographic skills and equipment proficiency.

Consistent with these observations, Ellis et al. [29] reported that only 41.2% of 435 physiotherapists used ultrasound as a procedural guide. This limited uptake may reflect its predominant association with electrotherapeutic interventions such as percutaneous electrolysis or peripheral nerve stimulation, rather than with dry needling per se. Although the role of ultrasound in physiotherapy has expanded substantially in recent years, its routine use for procedural guidance remains relatively uncommon. These findings reinforce the value of developing standardized, step-by-step protocols for blind procedures, particularly in anatomically sensitive regions, to improve clinical safety and technical consistency when ultrasound is unavailable.

### 4.5. Limitations and Future Directions

Several limitations must be acknowledged in this study. First, the preservation state of the cadaveric specimens likely influenced the results. Compared to in vivo conditions, increased tissue stiffness and dehydration may have altered the mechanical resistance of deep soft tissues, potentially affecting needle trajectory and redirection during insertion. Second, in some specimens, particularly those stored in a prone position, muscle mass displacement such as flattening of the gluteus maximus made palpation of key anatomical landmarks (e.g., the lateral edge of the ischial tuberosity and the posterior margin of the greater trochanter) more challenging. These alterations may have led to minor deviations in the intended entry point, thereby compromising procedural accuracy. Additionally, anatomical variability among specimens, including differences in age, sex, BMI, and degree of muscular atrophy, may have influenced clinical decisions regarding needle size selection and perceived depth, as well as the ease of identifying anatomical landmarks and achieving accurate placement. Third, the dry needling needles, which are low in diameter and easily deformable (0.30 mm), could have affected the accuracy due to bending during insertion. In the presence of increased tissue resistance, longer needles are more prone to deflection, especially when targeting deep anatomical planes. Another relevant aspect concerns the limited preparation time available to the physiotherapists before the procedure. The interventionists received the DDN protocol only 15 min prior to the intervention, without any opportunity for prior familiarization or rehearsal. Although both clinicians had experience with invasive procedures, the absence of prior training with the specific step-by-step protocol may have influenced their performance. The inclusion of only two physiotherapists limits the generalizability of the findings, which should be interpreted with caution when extrapolating to broader clinical practice. Additionally, intra-rater reliability assessments were only performed on a subset of cadavers, as repeated insertions were limited to specimens in better preservation condition. This selective repetition, driven by both anatomical viability and operational constraints, may have introduced a sampling bias, as intra-rater reproducibility could not be evaluated across the full set of specimens Moreover, the interpretation of agreement coefficients should be approached with caution due to the small sample size (n = 5), which may have led to inflated standard errors and wide confidence intervals across all variables (for example, the CI observed in SN.Punct.), thereby reducing the statistical robustness of the reliability estimates. However, bilateral procedures were performed on all five cadavers, resulting in ten complete interventions, which represents a relatively robust design compared to previous cadaveric studies that often included fewer specimens, as highlighted in recent systematic reviews [22]. Future studies should ensure systematic repetition across all specimens and evaluate the impact of protocol exposure time on procedural outcomes, particularly when comparing training effects or skill acquisition under different conditions.

### 4.6. Practical Applications

The findings of this study provide valuable insights for clinical and educational practice regarding the implementation of DDN in anatomically complex regions such as the deep gluteal space. While the procedure showed partial anatomical success and a moderate rate of sciatic nerve avoidance, nearly one-third of the procedures failed to reach the IFS, and nerve puncture occurred in almost one out of every six attempts. These results highlight the need for extreme caution in blind applications of this technique.

In settings where imaging is not available, the implementation of detailed, step-by-step protocols based on reliable anatomical landmarks and optimized needle direction, angle, and depth, may help reduce variability and enhance procedural safety [28]. These protocols may serve as a valuable educational tool, particularly for clinicians with limited experience in blind needling approaches.

Furthermore, the integration of dual validation through ultrasound and dissection, along with intra- and inter-rater reliability analysis, offers a methodological model for assessing procedural consistency and refining educational protocols [28]. Notably, the inclusion of an external observer to ensure adherence to the protocol strengthens the quality control of the intervention. This study illustrates how structured methodological designs can contribute to the development of safer and more accurate standardized approaches for invasive physiotherapy techniques, especially when applied to anatomically sensitive regions such as the deep gluteal space.

Given the 16.7% rate of sciatic nerve puncture observed, clinical application of this approach may require advanced anatomical knowledge and should ideally be supported by imaging when available. Additionally, the low inter-rater reliability in needle selection underscores the potential for variability in clinical decision-making, emphasizing the importance of enhanced training strategies, the use of anatomical reference tools, and structured protocols to promote consistent and safe implementation in clinical settings [22]. These findings are not intended to promote the routine clinical use of unguided deep dry needling, but to inform future research and training in anatomically complex regions.

## 5. Conclusions

A standardized, non-ultrasound-guided DDN protocol for the QF muscle demonstrated moderate anatomical accuracy and limited safety in cadaveric specimens. Despite the use of a step-by-step approach, 29.2% of the procedures failed to reach the IFS and 16.7% resulted in sciatic nerve puncture, indicating a relevant clinical risk. Intra- and inter-rater reliability was also limited, particularly for needle size selection, reflecting the complexity of consistent decision-making in blind procedures. The high agreement between ultrasound and dissection underscores the value of imaging validation for training, while detailed anatomical protocols remain essential when imaging is unavailable.

## Figures and Tables

**Figure 1 healthcare-13-01828-f001:**
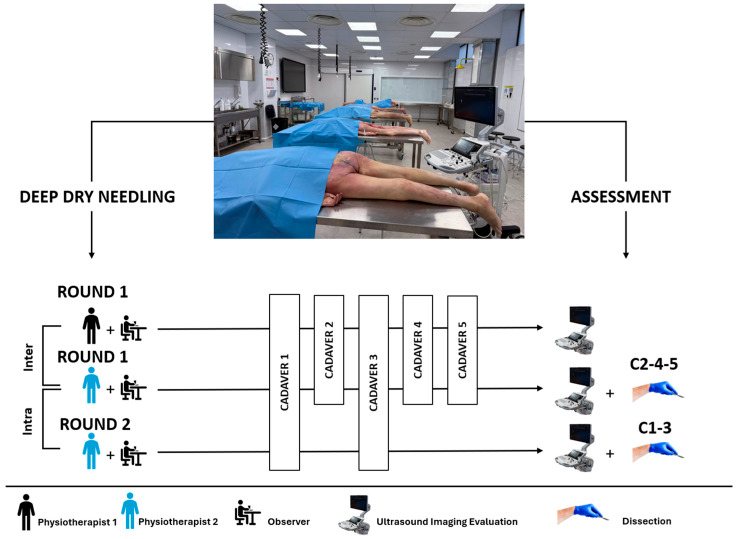
Diagram representing the methodology for assessing the reliability and accuracy of the approaches. Ph.1 (less experienced) performed a single round of bilateral approaches for each of the five cadaveric specimens. Ph.2 (10 years of experience) performed two rounds of bilateral approaches, puncturing all five specimens in the first round, while puncturing the first and third cadavers in the second round. The needles were fixed for the approaches of Ph.2 in the first round on cadavers 2, 3, and 5, and for cadavers 1 and 3 in the second round. All approaches were evaluated by ultrasound for both physiotherapists during their respective rounds. Abbreviations: Ph.1, physiotherapist 1; Ph.2, physiotherapist 2; R.1, first round of deep dry needling; R.2, second round of deep dry needling.

**Figure 2 healthcare-13-01828-f002:**
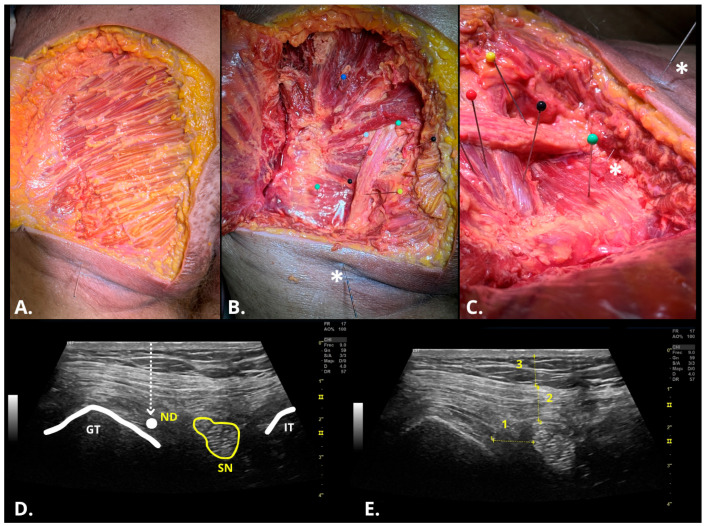
Cadaveric dissection and sonographic examination of the ischiofemoral space and quadratus femoris muscle. (**A**) Dissection of the superficial layer and subcutaneous tissue showing the gluteus maximus; (**B**,**C**) Dissection of the deep gluteal space and infrapiriform foramen showing the main muscular and vascular structures, as well as the deep dry needling needle (asterisk, *) (Blue: gluteus medius; Green: piriformis; Black: superior inferior gluteal artery; White: superior geminus; Orange: obturator internus; Black: inferior geminus; Yellow: sciatic nerve; Green: quadratus femoris); (**D**) Sonographic image of the ischiofemoral space with the sciatic nerve in the center of the image and the needle position after the out-of-plane approach; (**E**), Quantitative variables for ultrasound examination (1 = Distance from the needle tip to the sciatic nerve at the closest edge; 2 = Thickness of the gluteus maximus; 3 = Thickness of subcutaneous tissue). Abbreviations: GT, greater trochanter; IT, ischial tuberosity; ND, deep dry needling needle tip; SN, sciatic nerve.

**Table 1 healthcare-13-01828-t001:** Descriptive variables of the study cadavers.

Variables (n = 5)	Mean ± SD (Min.–Max.)	Median ± IQR (Min.–Max.)	95% CI				Shapiro–Wilk		
			Lower Bound	Upper Bound	Skewness	Kurtosis	Statistic	df	*p*-Value
Age, years	75.60 ± 14.10 (53.00–92.00)	78.00 ± 1.00 (53.00–92.00)	58.1	93.1	−1.06	2.59	0.86	4	0.239
Height, m	1.63 ± 5.22 (1.55–1.68)	1.65 ± 5.00 (1.55–1.68)	1.57	1.70	−1.31	1.40	0.89	4	0.347
Weight	71.00 ± 9.85 (55.00–80.00)	72.00 ± 8.00 (55.00–80.00)	58.8	83.2	−1.32	1.90	0.89	4	0.34
Estimated BMI	54.48 ± 4.30 (21.00–32.50)	26.40 ± 3.89 (21.00–32.50)	21.3	32	0.05	0.149	0.99	4	0.99
Hip Perimeter	97.80 ± 6.42 (90.00–107.00)	98.00 ± 6.00 (90.00–107)	89.8	106	0.410	0.172	0.99	4	0.964

Abbreviations: CI, confidence interval; df, degrees of freedom; IQR, interquartile range; SD, standard deviation.

**Table 2 healthcare-13-01828-t002:** Individual procedural data for each DDN approach performed on cadaveric specimens (n = 24).

Events (n = 24)	Nº (n = 5)	Side (L/R)	Physio. (n = 2)	Age (Years)	Height (m)	Weight (kg)	Sex (M/F)	Needle Size (mm)	Sub.Fat.Th (cm)	G.Max.Th (cm)	IFS (Y/N)	Success (Y/N)	SN. Punct. (Y/N)	SN.Dis. (cm)	Needle Depth (cm) (n = 14)
1	1a	R	1	92	1.55	78	F	0.30 × 50	0.85	1.24	No	No	No	1.29	–
2	1b	L	1	92	1.55	78	F	0.30 × 50	0.85	1.24	No	No	No	1.30	–
3	2a	R	1	77	1.65	72	M	0.30 × 40	1.00	1.27	Yes	No	No	1.88	–
4	2b	L	1	77	1.65	72	M	0.30 × 40	1.11	1.21	Yes	No	No	1.23	–
5	3a	R	1	78	1.68	70	M	0.30 × 40	0.94	1.36	Yes	Yes	No	1.86	–
6	3b	L	1	78	1.68	70	M	0.30 × 40	1.05	1.32	Yes	No	No	0.51	–
7	4a	R	1	53	1.62	55	M	0.30 × 40	0.50	1.11	Yes	No	No	1.51	–
8	4b	L	1	53	1.62	55	M	0.30 × 40	0.43	1.36	Yes	Yes	No	1.89	–
9	5a	R	1	78	1.67	80	M	0.30 × 40	1.16	0.86	Yes	Yes	No	3.23	–
10	5b	L	1	78	1.67	80	M	0.30 × 40	1.14	1.11	Yes	No	No	3.00	–
11	1a	R	2	92	1.55	78	F	0.30 × 60	1.22	1.75	Yes	Yes	No	1.78	1.85
12	1b	L	2	92	1.55	78	F	0.30 × 60	0.78	1.80	Yes	Yes	No	0.52	1.85
13	2a	R	2	77	1.65	72	M	0.30 × 60	0.76	0.92	Yes	No	Yes	0.00	5.10
14	2b	L	2	77	1.65	72	M	0.30 × 60	0.84	0.96	Yes	Yes	No	1.65	2.90
15	3a	R	2	78	1.68	70	M	0.30 × 60	0.89	1.67	Yes	Yes	No	0.80	4.20
16	3b	R	2	78	1.68	70	M	0.30 × 60	0.72	1.50	Yes	Yes	No	0.19	4.00
17	4a	R	2	53	1.62	55	M	0.30 × 30	0.46	0.95	Yes	No	Yes	0.00	2.10
18	4b	L	2	53	1.62	55	M	0.30 × 30	0.53	1.00	Yes	Yes	No	0.29	2.20
19	5a	R	2	78	1.67	80	M	0.30 × 60	0.95	0.94	Yes	Yes	No	0.21	3.50
20	5b	L	2	78	1.67	80	M	0.30 × 60	1.11	1.16	Yes	Yes	No	0.83	3.50
21	1a	R	2	92	1.55	78	F	0.30 × 60	0.74	1.44	Yes	Yes	No	1.13	5.00
22	1b	L	2	92	1.55	78	F	0.30 × 60	0.80	2.06	Yes	No	No	1.17	3.70
23	3a	R	2	78	1.68	70	M	0.30 × 60	0.70	0.79	Yes	Yes	No	1.31	2.00
24	3b	L	2	78	1.68	70	M	0.30 × 60	0.89	0.79	Yes	No	No	3.45	2.40

Abbreviations: Dis., distance; G.Max.Th., gluteus maximus thickness; IFS, ischiofemoral space; Punct., puncture; Sub.Fat.Th., subcutaneous fatty tissue thickness; SN, sciatic nerve; Succ., success.

**Table 3 healthcare-13-01828-t003:** Descriptive summary of anatomical, procedural and outcome variables across all interventions and stratified by examiner.

Variables		Mean ± SD [Min–Max) * or Frequency (%) †	
Physiotherapist	Ph.1: 10 (41.7%) †		
	Ph.2: 14 (58.3%) †		
Sex	Male	18 (75%) †	
	Female	6 (18%) †	
Needle Size	0.30 × 30:	2 (8.3%) †	Ph.1: 0 (0%) ‡
			Ph.2: 2 (100%) ‡
	0.30 × 40:	8 (33.3%) †	Ph.1: 8 (100%) ‡
			Ph.2: 0 (0%) ‡
	0.30 × 50:	2 (8.3%) †	Ph.1: 2 (100%) ‡
			Ph.2: 0 (0%) ‡
	0.30 × 60:	12 (50%) †	Ph.1: 0 (0%) ‡
			Ph.2: 12 (100%) ‡
IFS	Yes	22 (91.7%) †	Ph.1: 8 (36.4%) ‡
			Ph.2: 14 (63.6%) ‡
	No	(8.3%) †	Ph.1: 2 (100%) ‡
			Ph.2: 0 (0%) ‡
Success	Yes	13 (54.2%) †	Ph.1: 3 (23.1%) ‡
			Ph.2: 10 (76.9%) ‡
	No	13 (54.2%) †	Ph.1: 7 (63.7%) ‡
			Ph.2: 4 (36.3%) ‡
SN.Punct.	Yes	2 (8.3%) †	Ph.1: 0 (0%) ‡
			Ph.2: 2 (100%) ‡
	No	22 (91.7%) †	Ph.1: 10 (44.5%) ‡
			Ph.2: 12 (54.5%) ‡
Sub.Fat.Th (cm)		0.86 ± 0.22 (0.43–1.22) *	Ph.1: 0.90 ± 0.26 (0.43–1.16) *
			Ph.2: 0.81 ± 0.20 (0.46–1.22) *
G.Max.Th (cm)		1.24 ± 0.33 (0.79–2.06) *	Ph.1: 1.21 ± 0.15 (0.86–1.36) *
			Ph.2: 1.27 ± 0.43 (0.79–2.06) *
SN.Dis. (cm)		1.29 ± 0.96 (0.00–3.45) *	Ph.1: 1.77 ± 0.82 (0.51–3.23) *
			Ph.2: 0.95 ± 0.93 (0.00–3.45) *
Needle Depth (cm)		Ph.1: 3.16 ± 1.14 (1.85–5.10) *	

Abbreviations: Dis., distance; G.Max.Th., gluteus maximus thickness; IFS, ischiofemoral space; Punct., puncture; Sub.Fat.Th., subcutaneous fatty tissue thickness; SN, sciatic nerve; Succ., success. *** Mean and SD (Min.–Max.) were reported for quantitative variables with a normal distribution based on skewness, kurtosis and the Shapiro–Wilk test. † Frequency and percentage (%) were reported for qualitative variables using contingency tables. ‡ Subcategories additionally stratified by examiner. Values reflect the absolute frequency and percentage within each needle size category (not across total n).

**Table 4 healthcare-13-01828-t004:** Intra- and inter-rater reliability statistics on approaches to the quadratus femoris in cadavers.

Nominal Dichotomous Variables (Success/Failure)							
Variables	% Ph.1/R.1 Success/Failure (Yes/No) *		% Ph.2/R.2 Success/Failure (Yes/No) *	% Agreement	Cohen’s Kappa (κ) (95% CI)		N; *p*-Value
IFS Location (Y/N)	Intra-rater (Ph.2 vs. Ph.2)	100%	100%	100%	N/I ^†^		4; — ^‡^
	Inter-rater (Ph.1 vs. Ph.2)	80%	100%	80%	N/I ^†^		10; — ^‡^
SN. Punct. (Y/N) *	Intra-rater (Ph.2 vs. Ph.2)	0%	0%	100%	N/I ^†^		4; — ^‡^
	Inter-rater (Ph.1 vs. Ph.2)	20%	0%	80%	N/I ^†^		10; — ^‡^
Success (Y/N)	Intra-rater (Ph.2 vs. Ph.2)	100%	50%	50%	N/I ^†^		4; — ^‡^
	Inter-rater (Ph.1 vs. Ph.2)	30%	80%	50%	0.19 (−0.31–0.49)		10; 0.301
**Nominal Multicategorical Variables**							
**Nominal Variables**	**% Ph.1/R.1** **Needle Size (Fr)**		**% Ph.2/R.2** **Needle Size (Fr)**	**%** **Agreement**	**Mode** **Frequency (%)**	**Cohen’s Kappa (κ) (95% CI)**	**N; *p*-Value**
Needle Size	Intra-rater (Ph.2 vs. Ph.2)	0.30 × 40 mm (2)	0.30 × 60mm (2)	50%	50%	N/I ^†^	4; — ^‡^
	Inter-rater (Ph.1 vs. Ph.2)	0.30 × 40 mm (8)	0.30 × 60 mm (6)	0%	60%	N/I ^†^	10; — ^‡^

Abbreviations: CI, confidence interval; IFS, ischiofemoral space location; N, number of observations; N/I, not interpretable; Ph., physiotherapist; R., rater; SN.Punct., sciatic nerve puncture; Ph.1 = Physiotherapist 1/Rater 1 (clinical experience < 10 years); Ph.2 = Physiotherapist 2/Rater 2 (clinical experience > 10 years). * The success rating for sciatic nerve puncture was determined according to whether the needle contacted the nerve (classified as failure, i.e., ‘yes’ = nerve puncture = failed approach) or avoided it (classified as success, i.e., ‘no’ = no puncture = successful approach). † Cohen’s kappa was equal to 0.000 due to low or null variability in one of the rating distributions and the value was not interpretable. ‡ Cohen’s kappa was equal to 0.000 due to a complete lack of agreement among raters. The corresponding *p*-value would be 1.000 under the null hypothesis of chance agreement.

**Table 5 healthcare-13-01828-t005:** Agreement between ultrasound and dissection methods for dry needling outcomes in cadavers.

Variables (n = 10)	US Positive (%)	Dis. Positive (%)	% Agreement	Valence Index	Cohen’s Kappa (95% CI)	N; *p*-Value
IFS Location	100%	100%	100%	1.00	N/I ^†^	10; — ^‡^
SN.Punct.	20%	10%	90%	0.80	0.62 (0.04–1.28)	10; **0.035**
Success	60%	40%	100%	0.20	1.00 (1.00–1.00)	10; **0.002**

Abbreviations: CI, confidence interval; Dis., cadaveric dissection; IFS, ischiofemoral space location; N/I, not interpretable; SN.Punct., sciatic nerve puncture; US, ultrasound imaging. Note: Agreement values were computed as the proportion of matching YES/YES or NO/NO classifications across modalities. Cohen’s kappa was calculated to determine the intermethod agreement adjusted for chance. Valence Index reflects the absolute difference between the proportion of positive and negative outcomes. † Cohen’s kappa was equal to 0.000 due to low or null variability in one of the rating distributions and the value was not interpretable. ‡ Cohen’s kappa was equal to 0.000 due to a complete lack of agreement among raters. The corresponding *p*-value would be 1.000 under the null hypothesis of chance agreement. Statistically significant agreement between ultrasound and dissection is highlighted (**bold**) for those cases where *p* < 0.05.

## Data Availability

The raw data supporting the conclusions of this article will be made available by the authors on request.

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
