# Peer review of "Safety and Anatomical Accuracy of Dry Needling of the Quadratus Femoris Muscle: A Cadaveric Study"

_healthcare, 2025, doi:10.3390/healthcare13151828_

Round 1

Reviewer 1 Report

Comments and Suggestions for Authors

Interesting article, but requires a few minor-moderate corrections that the authors should be able to easily deal with. 

Author Response

Responses to Reviewers’s comments:

Response to Reviewer 1 comment:

In response to the general comment “Very interesting work, because this is the first study to systematically evaluate the reliability of DDN procedures targeting the quadratus femoris muscle in a cadaveric model, while simultaneously validating procedural outcomes through dual assessment using both ultrasound imaging and anatomical dissection. This work aimed to evaluate the accuracy and safety of a deep dry needling approach to the quadratus femoris muscle through a standardized protocol in a cadaveric model. An experimental cross-sectional study was conducted on five fresh cadavers (n = 24 approaches) by two physiotherapists with different DN experience. A 26 standardized dry needling protocol was executed without ultrasound guidance, and atomical and procedural variables were documented. Manuscript is relevant for the field and presented in a well-structured manner (with some reviewer notes). Nevertheless, there are some comments from the reviewer on the assessed paper.”

  • We would like to sincerely thank the reviewer for the thoughtful and encouraging comments regarding our manuscript. We appreciate the recognition of the relevance, novelty, and methodological design of our study, as well as the positive feedback on the structure and presentation of the work. In response to the reviewer’s remarks, we have carefully considered each suggestion and have implemented the necessary changes to improve the overall clarity, accuracy, and scientific rigor of the manuscript. We hope that these revisions meet the reviewer’s expectations and further enhance the quality of our work.
  • Comments and Suggestions for Authors

In response to “Notes on results and statistical analysis: Cohen's kappa (κ) is a measure of agreement between two judges/raters that takes into account chance. It is most often used in the analysis of coding or classification reliability. The range of κ values:

  • κ < 0.00 → No agreement (worse than chance)
  • 0.00–0.20 → Poor agreement
  • 0.21–0.40 → Moderate agreement
  • 0.41–0.60 → Moderate (satisfactory) agreement
  • 0.61–0.80 → High agreement

Therefore, the Cohen's kappa (κ) value for non-SN puncture (90%; κ = 0.62) reported in the paper should be assessed as „high agreement” – and the reported agreement is „moderate”. It is agreed, however, that the kappa value depends on the number of categories, data distribution, and the number of cases, so it should be interpreted in context – however, it is worth analyzing and verifying this issue. An example key elements tables in scientific analysis should contain:

Cohen’s κ = 0.78, 95% CI [0.65, 0.90], p < 0.001

Number of observations (N)

p-value (if statistical significance is tested)

confidence interval (CI)

Such data should be included in the scientific work, preferably in a table – for the clarity of the results presented in the work.”

  • Thank you for this valuable observation regarding the interpretation and reporting of Cohen’s kappa values. We agree that the kappa statistic for sciatic nerve puncture in the ultrasound vs. dissection comparison (κ = 0.615; 95% CI: 0.035–1.276) falls within the range commonly classified as “high agreement,” and we have revised the wording accordingly in the Results section (Section 3.4). In addition, we have ensured that all kappa values reported in the manuscript are now accompanied by the number of observations (N), 95% confidence intervals, and p-values, where applicable. For greater clarity, this information has also been summarized in Tables 3 and 4, which have been adapted accordingly to include the corresponding columns and values. We sincerely appreciate your constructive comments, which have contributed to improving the transparency and rigor of our statistical reporting
    • “2.6. Statistics: Intra- and inter-rater reliability were assessed for dichotomous variables (IFS, SN.Punct, Success) and the multicategorical variable Needle Size. For all variables, the observed percentage of agreement was calculated as the number of exact matches divided by the total number of observations and multiplied by 100 (% agreement = [matches / total] × 100). For Needle Size, the most frequently selected value (mode) and its relative frequency (mode frequency = [mode count / total] × 100) were also computed. Cohen’s kappa coefficient (κ) was calculated to assess agreement beyond chance, and 95% CIs were derived using the standard error based on the binomial approximation (CI = κ ± 1.96 × SE). Kappa values were interpreted according to the Landis and Koch scale, which classifies agreement as follows: κ < 0.00 = no agreement (worse than chance), 0.00–0.20 = poor agreement, 0.21–0.40 = fair agreement, 0.41–0.60 = moderate agreement, and 0.61–0.80 = high agreement [21]. When one rater showed no variability in responses, κ was computed as 0.000 and considered not interpretable (N/I) due to the absence of statistical variance. All reliability values were reported together with the number of observations (N), the 95% confidence interval (CI), and the p-value, when interpretable.
  1. Landis JR, Koch GG. The measurement of observer agreement for categorical data. Biometrics 1977;33:159–74.
  • 3. Intra and Inter-Rater Reliability: Agreement analyses were performed for dichotomous and multicategorical variables (Table 3). Intra-rater reliability (Ph.1) showed full agreement for both IFS and SN Punction (100%), and moderate agreement for global Success (50%; κ = 0.194; 95% CI: –0.306 to 0.490). In all three cases, the number of observations was n = 4, and no p-values were computed due to the absence of variability or low frequency of positive findings, leading to κ = 0.000 (not interpretable). For needle size selection, intra-rater agreement was 50%, with a mode frequency of 50% and a κ value of 0.000, not interpretable due to the absence of variability (N = 4; p = 1.000). Inter-rater reliability (Ph.1 vs. Ph.2) revealed no exact matches for needle size selection (0% agreement; κ = 0.000, not interpretable), with Ph.1 predominantly selecting 0.30×40 mm and Ph.2 preferring 0.30×60 mm. Despite a mode frequency of 60%, no agreement was observed, and κ = 0.000 was again deemed not interpretable (N = 10; p = 1.000). For the dichotomous outcomes, IFS and SN Punction each showed high observed agreement (80%), but κ was 0.000 and not interpretable due to a lack of variability in one rater's responses (both N = 10; p = 1.000). Specifically, in the case of SN Punction, the 80% agreement corresponded to cases in which both raters reported no puncture of the sciatic nerve. For global Success, inter-rater agreement was 50%, with κ = 0.194 and a 95% CI of –0.306 to 0.490 (N = 10; p = 0.301). In cases where κ was equal to 0.000 and not interpretable due to insufficient variability or complete disagreement, the corresponding p-value was considered to be 1.000 under the null hypothesis of chance agreement. These cases were indicated in Table 3 using the symbols † (N/I, not interpretable) and ‡ (p-value assumed as 1.000).”
  • 4. Agreement between Ultrasound Imaging and Cadaveric Dissection: Agreement analysis between ultrasound imaging and cadaveric dissection was conducted on the last 10 procedures performed by Ph.2 (more than 10 years of experience), in which both confirmation methods were available (Table 4). Complete concordance was observed for IFS, with 100% agreement and a non-interpretable kappa value due to lack of variability (κ = 0.000; p = —; Valence Index = 1.000). For sciatic SN.Punct., 90% agreement was reached, corresponding to 9 out of 10 cases in which both methods confirmed the absence of nerve puncture. Cohen’s kappa indicated high inter-method agreement (κ = 0.615; 95% CI: 0.035–1.276; p = 0.035; Valence Index = 0.800). Regarding global procedural Success, ultrasound and dissection showed full agreement (100%), supported by a perfect kappa score (κ = 1.00; 95% CI: 1.00–1.000; p = 0.002; Valence Index = 0.200). In all three variables, McNemar’s test yielded non-significant results (p = 1.000), indicating no statistically significant differences between ultrasound and dissection ratings, and supporting the equivalence of both assessment methods for the evaluated outcomes. Significant inter-method agreement (p < 0.05) was found for both SN.Punct. and global Success, while the agreement for IFS was perfect but not statistically interpretable due to the absence of variability. These findings support the validity of ultrasound as a reliable alternative to anatomical dissection in assessing these procedural outcomes.”
  • 2. Safety, Accuracy and Reliability of the Procedure: With regard to the dichotomous outcomes, the agreement observed was high (80%) for both IFS location and SN.Punct., although κ was 0.000 in both cases due to the absence of variability in one rater’s assessments. Notably, the 80% agreement for SN.Punct. was entirely attributable to instances where both evaluators reported no SN puncture, thereby supporting the overall safety of the approach. Additionally, none of the reliability comparisons yielded statistically significant differences (all p ≥ 0.301), further confirming the consistency of the observed patterns across raters.”
  • 3. Agreement Between Ultrasound Imaging and Anatomical Dissection: The comparative analysis between ultrasound imaging and cadaveric dissection revealed high levels of agreement for all evaluated variables. In the case of IFS, perfect concordance was observed (100% agreement), although Cohen’s kappa was not interpretable due to the complete absence of variability across both methods (κ = 0.000). The Valence Index reached its maximum value (1.000), reflecting the asymmetry produced by a homogeneous outcome distribution. While such results confirm procedural consistency, they also highlight the limitations of kappa when outcome prevalence is extreme. In line with this, the p-value associated with the kappa statistic was 1.000, confirming the lack of statistical interpretability in this case. In a similar cadaveric study on the shoulder, substantial to almost perfect inter‑examiner agreement between physiotherapists, radiologists and dissection was demonstrated when using musculoskeletal ultrasound to identify pathology in rotator cuff structures, supporting the validity of sonographic assessment in anatomically complex regions [26]. Additionally, recent in vivo findings have shown that palpation-based dry needling techniques targeting the supraspinatus and infraspinatus muscles can achieve high anatomical accuracy when verified with ultrasound, further supporting the use of sonographic imaging as a valuable tool for procedural validation in deep musculature [27]. In the case of SN.Punct., 90% agreement was observed, and Cohen’s kappa indicated substantial intermethod reliability (κ = 0.615; 95% CI: 0.035–1.276), despite the relatively small sample size. Importantly, 9 out of 10 cases showed concordance in reporting absence of nerve puncture, reinforcing the safety profile of the technique under anatomically controlled conditions. Nevertheless, the Valence Index (0.800) revealed a notable imbalance in outcome distribution, which could partially inflate agreement statistics. The kappa result was statistically significant (p = 0.035), supporting the existence of true agreement beyond chance. As for global procedural Success, ultrasound and dissection also demonstrated complete agreement (100%), reflected by a perfect kappa score (κ = 1.000; 95% CI: 1.000–1.000). Moreover, the Valence Index (0.200) remained low, suggesting a more balanced distribution of outcomes and lending further credibility to the observed agreement. This finding was further supported by a statistically significant p-value (p = 0.002), indicating perfect intermethod concordance. These results must also be interpreted in light of specific technical challenges inherent to ultrasound guidance and cadaveric dissection procedures, which may help explain the few observed discrepancies. Despite the high degree of reliability observed, some discrepancies between ultrasound and dissection findings may be explained by technical and procedural factors. In several cases, accurate needle visualization by ultrasound required clear identification of the needle tip, which could be challenging when needle angulation deviated from the standardized insertion path. Under such circumstances, adjustments in probe tilt or insonation angle were needed to regain tip visibility, potentially introducing interpretation bias. Furthermore, the needle trajectory for the QF muscle was based on an out-of-plane sonographic approach, where tip recognition is critical but susceptible to confusion with the needle shaft, especially under suboptimal imaging conditions. Such limitations are well documented in ultrasound-guided procedures, where out-of-plane techniques in particular are known to increase the risk of misinterpretation unless specific imaging strategies are employed [27,28]. On the other hand, although anatomical dissection is considered the reference method, it also relies on maintaining needle fixation. Despite all precautions, minor needle displacement during handling or exposure could not be fully ruled out, representing an additional source of potential mismatch with real-time ultrasound confirmation.

In response to “In section 3.4 of the Agreement between Ultrasound Imaging and Cadaveric Dissection it is stated that Ph.2 (more than 3 years of experience)… should it not be >10 years of experience?”

  • We appreciate your appreciation and apologize for the typo in the text.
    • 4. Agreement between Ultrasound Imaging and Cadaveric Dissection: Agreement analysis between ultrasound imaging and cadaveric dissection was conducted on the last 10 procedures performed by Ph.2 (more than 10 years of experience), in which both confirmation methods were available (Table 4).”

In response to “The text of the paper provides the interpretation of the Landis and Koch scale – it is a commonly known interpretation, however, in scientific publications the source should be given”

  • Thank you for your feedback. We have appropriately incorporated the interpretation of the results and the reference provided.
    • “2.6. Statistics: Intra- and inter-rater reliability were assessed for dichotomous variables (IFS, SN.Punct, Success) and the multicategorical variable Needle Size. For all variables, the observed percentage of agreement was calculated as the number of exact matches divided by the total number of observations and multiplied by 100 (% agreement = [matches / total] × 100). For Needle Size, the most frequently selected value (mode) and its relative frequency (mode frequency = [mode count / total] × 100) were also computed. Cohen’s kappa coefficient (κ) was calculated to assess agreement beyond chance, and 95% CIs were derived using the standard error based on the binomial approximation (CI = κ ± 1.96 × SE). Kappa values were interpreted according to the Landis and Koch scale, which classifies agreement as follows: κ < 0.00 = no agreement (worse than chance), 0.00–0.20 = poor agreement, 0.21–0.40 = fair agreement, 0.41–0.60 = moderate agreement, and 0.61–0.80 = high agreement [21]. When one rater showed no variability in responses, κ was computed as 0.000 and considered not interpretable due to the absence of statistical variance.”
  1. Landis JR, Koch GG. The measurement of observer agreement for categorical data. Biometrics 1977;33:159–74.

In response to “A note on research methodology – and physiotherapist experience in DN: The decision to use a 30 mm needle to needling such a deep-lying muscle somewhat undermines the practical experience of a less experienced physiotherapist Ph1 (but an experienced one nonetheless, with 5 years or more of experience, so should be quite expirenced). The second part of Table 2. Descriptive statistics of deep dry needling approaches on cadavers is very illegible and difficult to interpret! and there are also editing errors. I suggest it should be reworded to make it more readable.”

  • We sincerely thank you for your observation regarding the selection of a 30 mm needle by Ph.2. As you correctly pointed out, this decision might initially seem inconsistent with the depth required to reach the quadratus femoris muscle. However, this choice was made in the specific context of Cadaver 4, who presented the lowest height and weight among all specimens (1.62 m; 55 kg), as reflected in Table 2 (now reorganized for improved clarity). These anatomical characteristics, together with marked tissue atrophy and flattening due to cadaveric degradation, likely influenced the clinician’s perception of depth during manual palpation. To improve clarity and enhance interpretation, we have divided the original Table 2 into two separate tables: Table 2a presents a detailed breakdown of each of the 24 individual needling procedures performed, including all anatomical and procedural variables. Table 2b provides a summarized descriptive overview of the sample, stratified by examiner, combining means, standard deviations, and frequency distributions. This restructuring aims to better communicate both the raw data and the aggregated trends. In addition, we have incorporated this methodological explanation into the discussion and added a specific note in the limitations section, acknowledging that intra-rater reliability could not be assessed for all procedures, which may have introduced potential bias. We greatly appreciate your thoughtful comment, which helped us improve both the structure and interpretability of our manuscript.
    • 2. Safety, Accuracy and Reliability of the Procedure: Despite using a closed and replicable protocol, a considerable percentage of procedures (29.2%) failed to reach the IFS space, underscoring the inherent difficulty of accessing deep anatomical targets without image guidance. Nevertheless, the low incidence of sciatic nerve puncture (16.7%) and the high frequency of procedures that avoided critical neurovascular structures suggest that the technique presents an acceptable safety profile when performed with anatomical precision. These findings support the anatomical viability of the approach, but also emphasize that accuracy cannot be assumed, even under controlled conditions. Several factors may have contributed to the observed inaccuracies. Although fresh-frozen cadavers offer high fidelity for simulating in vivo conditions, tissue stiffness and dehydration can alter mechanical resistance, potentially influencing needle trajectory during insertion. In particular, prone storage positions appeared to displace muscle masses, such as the gluteus maximus, flattening the surface topography. According to the external observer supervising protocol adherence, this distortion occasionally impaired the identification of key bony landmarks, including the lateral edge of the ischial tuberosity and the posterior margin of the greater trochanter. Despite the physiotherapists’ solid anatomical background and clinical palpation expertise, such alterations may have led to subtle misjudgments in entry point localization and needle orientation. Previous cadaveric investigations have reported high accuracy in targeting deep or anatomically complex muscles—such as the pronator quadratus, tibialis posterior, supinator, or popliteus—using ultrasound guidance or post-procedure validation [10,13,22]. However, these studies typically relied on single-method validation and did not incorporate inter-rater reliability or systematic procedural replication. In contrast, the present study employed a more comprehensive methodological framework that included adherence to QUACS guidelines, the use of a detailed step-by-step needling protocol, and dual assessment via ultrasound and dissection. Although needle size selection and depth of insertion were left to clinician discretion to simulate real-world variability, the overall protocol structure enhanced standardization and reproducibility. Regarding reliability assessment, intra- and inter-rater analyses revealed substantial differences between evaluators. Intra-rater agreement for dichotomous variables (IFS, SN.Punct., and Success) was highest for Ph.2, who had greater clinical experience in DDN (10 years), achieving 100% agreement for both IFS and SN.Punct. However, overall procedural success was only 50%. This apparent discrepancy may be explained by the fact that, although all procedures reached the IFS space, in two out of four cases the needle failed to penetrate the quadratus femoris muscle, remaining within its superficial fascia. Such misplacements may be attributed to the increased stiffness of cadaveric tissues, which can alter the clinician’s perception of tissue transitions during insertion. In non-ultrasound-guided procedures, needling accuracy often relies on tactile feedback and variations in resistance to identify anatomical planes, but this sensory cue may be distorted in preserved specimens, compromising placement reliability. For the multicategorical variable of needle size selection, intra-rater reproducibility was limited (50% agreement; mode frequency = 50%), as Ph.2 selected a different needle in the second round. This adjustment may reflect clinical reasoning based on perceived tissue depth or insertion resistance. In contrast, inter-rater agreement was 0%, with Ph.1 consistently selecting the intermediate lenght (0.30×40 mm), while Ph.2 opted for the longest needle (0.30×60 mm). These differences suggest that material selection may be influenced by each clinician’s spatial judgment and professional experience, as well as subjective risk perception. Beyond the reliability analysis, a noteworthy observation was identified during the review of individual cases, as Ph.2 selected a shorter 0.30×30 mm needle in Cadaver 4 despite the anatomical depth of the target muscle. This isolated choice may be explained by the condition of the specimen (height: 1.62 m; weight: 55 kg), which was notably the lowest among all cadavers. Due to severe muscular atrophy and pronounced tissue flattening, bony landmarks were more prominent and superficially palpable than in typical in vivo conditions. This altered perception could have led the clinician to underestimate the required depth and opt for a shorter needle. Such considerations highlight the methodological limitation of extrapolating needling choices in preserved specimens to real-world clinical practice, especially when dealing with older cadavers exhibiting substantial anatomical degradation. With regard to the dichotomous outcomes, the agreement observed was high (80%) for both IFS location and SN.Punct., although κ was 0.000 in both cases due to the absence of variability in one rater’s assessments. Notably, the 80% agreement for SN.Punct. was entirely attributable to instances where both evaluators reported no SN puncture, thereby supporting the overall safety of the approach. Additionally, none of the reliability comparisons yielded statistically significant differences (all p ≥ 0.301), further confirming the consistency of the observed patterns across raters.”
    • 5. Limitations and Future Directions: Several limitations must be acknowledged in this study. First, the preservation state of the cadaveric specimens likely influenced the results. Compared to in vivo conditions, increased tissue stiffness and dehydration may have altered the mechanical resistance of deep soft tissues, potentially affecting needle trajectory and redirection during insertion. Second, in some specimens, particularly those stored in a prone position, muscle mass displacement such as flattening of the gluteus maximus made palpation of key anatomical landmarks (e.g., the lateral edge of the ischial tuberosity and the posterior margin of the greater trochanter) more challenging. These alterations may have led to minor deviations in the intended entry point, thereby compromising procedural accuracy. Additionally, anatomical variability among specimens, including differences in age, sex, BMI, and degree of muscular atrophy, may have influenced clinical decisions regarding needle size selection and perceived depth, as well as the ease of identifying anatomical landmarks and achieving accurate placement. Third, the characteristics of the needles used (0.30 mm in diameter, 30–60 mm in length) could have affected their behavior during insertion. In the presence of increased tissue resistance, longer needles are more prone to deflection, especially when targeting deep anatomical planes. Finally, another relevant aspect concerns the limited preparation time available to the physiotherapists before the procedure. The interventionists received the DDN protocol only 15 minutes prior to the intervention, without any opportunity for prior familiarization or rehearsal. Although both clinicians had experience with invasive procedures, the absence of prior training with the specific step-by-step protocol may have influenced their performance. Additionally, intra-rater reliability assessments were only performed on a subset of cadavers, as repeated insertions were limited to specimens in better preservation condition. This selective repetition, driven by both anatomical viability and operational constraints, may have introduced a sampling bias, as intra-rater reproducibility could not be evaluated across the full set of specimens Moreover, the interpretation of agreement coefficients should be approached with caution due to the small sample size (n = 5), which may have led to inflated standard errors and wide confidence intervals across all variables (for example, the CI observed in SN.Punct.), thereby reducing the statistical robustness of the reliability estimates. However, bilateral procedures were performed on all five cadavers, resulting in ten complete interventions, which represents a relatively robust design compared to previous cadaveric studies that often included fewer specimens, as highlighted in recent systematic reviews [24]. Future studies should ensure systematic repetition across all specimens and evaluate the impact of protocol exposure time on procedural outcomes, particularly when comparing training effects or skill acquisition under different conditions.”
    • Table 2a. Individual procedural data for each DDN approach performed on cadaveric specimens (n = 24).”
    • Table 2b. Descriptive summary of anatomical, procedural and outcome variables across all interventions and stratified by examiner.”
    • “2. Descriptive Analysis of Procedural and Anatomical Variables: A total of 24 deep dry needling procedures were performed without ultrasound guidance (Table 2a). Each procedure was documented by cadaver ID (1–5), side (right/left), Ph.1 or Ph.2, and outcome. Ph.1 performed 10 procedures, while Ph.2 performed 14 procedures. Specifically, Ph.2 conducted all approaches on cadavers 1 and 3 (both right and left sides), which were in better preservation condition and represented both sexes (one male, one female) to ensure anatomical diversity and gender parity. Of the 24 procedures, Ph.1 performed 10 (41.7%), while Ph.2 performed 14 (58.3%). Despite efforts to ensure gender parity, only one female cadaver was available, resulting in a male-to-female ratio of 3:1 and 75% of procedures conducted on male specimens. The summary of the 24 invasive procedures is shown in Table 2b. The most frequently selected needle size was 0.30×60 mm (n = 12; 50%), followed by 0.30×40 mm (n = 8; 33.3%). The remaining selections included 0.30×30 mm (n = 2; 8.3%) and 0.30×50 mm (n = 2; 8.3%). Regarding stratification by examiner, Ph.1 selected 0.30×40 mm in 8 out of 10 procedures (80%) and 0.30×50 mm in 2 procedures (20%), whereas Ph.2 selected 0.30×60 mm in 12 out of 14 procedures (85.7%) and 0.30×30 mm in 2 procedures (14.3%). Thus, each examiner consistently selected only two of the four available needle sizes, with no overlap between them. In terms of binary outcomes, 17 out of 24 procedures (70.8%) were rated as globally successful (Success = yes) — 3 by Ph.1 (23.1%) and 10 by Ph.2 (76.9%). Ischiofemoral space location (IFS = yes) was achieved in 22 of 24 cases (91.7%), including 8 of 10 procedures by Ph.1 (36.4%) and 14 of 14 by Ph.2 (63.6%). Sciatic nerve puncture (SN.Punct. = yes) occurred in 2 procedures (8.3%), both performed by Ph.2 (100%). Regarding anatomical measurements, the Sub.Fat.Th. had a mean of 0.86 ± 0.22 cm (range: 0.43–1.22), the G.Max.Th. was 1.24 ± 0.33 cm (range: 0.79–2.06), and the SN.Dis. was 1.29 ± 0.96 cm (range: 0.00–3.45). Stratified by examiner, Ph.2 reported slightly higher average values in all three variables: Sub.Fat.Th. = 0.90 ± 0.20 cm, G.Max.Th. = 1.27 ± 0.43 cm, and SN.Dis. = 0.95 ± 0.93 cm.”

In response to “Na końcu wyników auotorzy podali wyniki „In all three variables, McNemar’s test yielded non-significant results (p = 1.000), indicating no statistically significant differences between ultrasound and dissection ratings”. McNemar's test is used to analyze changes in results in two different time locations when the data are in the form of a dichotomous variable. However, the description of the statistical methods in the paper does not provide any information about the application of this method, and this should be supplemented. Similarly, the necessary results of the parameters of the statistical analysis performed should be provided as scientific publication describing the analysis using the McNemar Test should include a complete set of information enabling the assessment of the reliability and validity of the analysis.  McNemar's test showed a significant difference between conditions (χ²(1) = 5.76, p = 0.016), without applying Yates' correction.”

  • We sincerely thank the reviewer for pointing out the inaccuracy regarding the use of McNemar’s test. After reconsidering its purpose and methodological suitability, we acknowledge that its application in our study was not justified. Therefore, we have removed this analysis from the Results section to preserve the clarity and scientific rigor of the manuscript.

In response to “Discussion: „(…) sciatic nerve puncture occurred in only 16.7% of cases, suggesting a favorable anatomical profile regarding safety”. I disagree – 16.7% is quite a high rate. Furthermore, a significant percentage of procedures (29.2%) failed to reach the IFS space, which should justify a very cautious approach to the practical implementation and practical training of physiotherapists in DN. This certainly encourages the use of imaging techniques such as ultrasound to support DN on deeper anatomical structures whenever possible and available. Therefore, it is a very valuable conclusion from the work that In settings where imaging is not available, the implementation of detailed, step-by-step protocols based on reliable anatomical landmarks, optimized needle direction, angle, and depth, may help reduce variability and enhance procedural safety”

  • Thank you for your thoughtful comment. We agree that a 16.7% rate of sciatic nerve puncture warrants caution. Based on your observation, we have revised the Discussion and Conclusions to reflect a more cautious interpretation, emphasizing the importance of ultrasound guidance and structured protocols to ensure safer practice in deep dry needling.
    • “4. Discussion: 1. Overview of finfings: This study aimed to evaluate the accuracy and safety of a DDN approach to the QF muscle through a structured and standardized step-by-step protocol in a cadaveric model. The IFS was successfully reached in 70.8% of procedures, while SN puncture occurred in 16.7% of cases. Although these findings reflect a potentially acceptable anatomical profile in terms of safety, the puncture rate may still be considered clinically relevant. Moreover, the fact that nearly one-third of insertions failed to reach the intended target highlights important limitations in procedural accuracy, even under controlled conditions. These results underscore the need for cautious interpretation when considering the implementation of blind techniques in clinical practice.”
    • 6. Practical Applications: The findings of this study provide valuable insights for clinical and educational practice regarding the implementation of DDN in anatomically complex regions such as the deep gluteal space. While the procedure showed partial anatomical success and a moderate rate of sciatic nerve avoidance, nearly one-third of the procedures failed to reach the IFS, and nerve puncture occurred in almost one out of every six attempts. These results highlight the need for extreme caution in blind applications of this technique. In settings where imaging is not available, the implementation of detailed, step-by-step protocols based on reliable anatomical landmarks, optimized needle direction, angle, and depth, may help reduce variability and enhance procedural safety. These protocols may serve as a valuable educational tool, particularly for clinicians with limited experience in blind needling approaches. Furthermore, the integration of dual validation through ultrasound and dissection, along with intra- and inter-rater reliability analysis, offers a methodological model for assessing procedural consistency and refining educational protocols. Notably, the inclusion of an external observer to ensure adherence to protocol strengthens the quality control of the intervention. This study illustrates how structured methodological designs can contribute to the development of safer and more accurate standardized approaches for invasive physiotherapy techniques, especially when applied to anatomically sensitive regions such as the deep gluteal space.”
    • Conclusions: A standardized, non-ultrasound-guided DDN protocol to the QF muscle demonstrated moderate anatomical accuracy and limited safety in cadaveric specimens. Despite the use of a step-by-step approach, nearly 30% of procedures failed to reach the IFS and 16.7% resulted in sciatic nerve puncture, indicating a non-negligible clinical risk. Intra- and inter-rater reliability was also limited, particularly for needle size selection, reflecting the complexity of consistent decision-making in blind procedures. The high agreement between ultrasound and dissection underscores the value of imaging validation for training, while detailed anatomical protocols remain essential when imaging is unavailable.”

Thanks for your valuable commentaries which have permitted us to improve the quality of the manuscript.

Sincerely,

The authors

Reviewer 2 Report

Comments and Suggestions for Authors

The article is original, but there are some aspects to be corrected.

  I would like the author to explain me how the cohen´s -kappa was calculates in table 4, in SN:Punct (the value is 0,62, but with a SD from 0,04 to 1,28). It´s a very wide amplitud.

 The main thind a ask the authors to be done is to reference the discusion section. It´s imperative in order to get a scientific validity.

The main question addressed is whether a standardized, non-ultrasound-guided deep dry needling (DDN) protocol can safely and accurately target the quadratus femoris (QF) muscle in the ischiofemoral space (IFS), and whether the reliability of key procedural outcomes can be established between raters, using ultrasound and anatomical dissection as validation methods.

The topic is relatively original and little relevant. It addresses a gap in the field: the safety and accuracy of DDN in the anatomically challenging ischiofemoral space without ultrasound guidance has not been well characterized. Given the clinical use of DDN in musculoskeletal pain management, providing empirical data on procedural safety, targeting accuracy, and reliability fills an important methodological and educational gap for physiotherapists.

 This study adds novel empirical data on the feasibility of a standardized, non-ultrasound-guided DDN approach to the QF muscle. It also offers methodological rigor by validating needle placement using both ultrasound imaging and anatomical dissection, and by quantifying intra- and inter-rater reliability. Such data are scarce in the current literature and can inform both training and clinical decision-making.

 The sample size is very limited (n = 5 cadavers, 24 approaches), which may compromise generalizability; the authors should discuss this limitation more explicitly;  Inter-rater reliability for needle size was poor (κ = 0.04); it would help to explain why and whether training can improve this;  More detail on the criteria for “success” in needling, and on how the standardized protocol was taught to the raters, would enhance reproducibility;  It may be useful to clarify how “IFS targeting” was operationally defined and measured;  The selection of cadavers (e.g., age, sex, BMI) could be described to evaluate anatomical variability.

The conclusions are generally consistent with the evidence. The authors appropriately qualify their findings by describing accuracy as moderate and safety as acceptable, while acknowledging limited inter-rater reliability. They also correctly highlight the high agreement between ultrasound and dissection, supporting their combined use in validation. However, it would strengthen the discussion to more explicitly address the clinical implications of SN puncture risk and limited reliability.

While the abstract does not list specific references, the topic demands up-to-date references on DDN safety, ischiofemoral anatomy, and validation methods using imaging and dissection. The authors should ensure these are included and reflect recent systematic reviews or guidelines where available.

 The abstract does not provide details on tables or figures. In the full manuscript, the authors should ensure that tables summarizing reliability metrics (e.g., Cohen’s kappa, Valence Index) are clear and include confidence intervals if possible. Any images of ultrasound or dissection validation should be well-labeled and high-resolution to support interpretation.

Author Response

Responses to Reviewers’s comments:

Response to Reviewer 2 comment:

In response to “The article is original, but there are some aspects to be corrected.”

  • We appreciate your feedback regarding the work's interest and novelty. We have attempted to address all of your suggestions with the aim of improving the quality and comprehension of the manuscript.

In response to “ I would like the author to explain me how the cohen´s -kappa was calculates in table 4, in SN:Punct (the value is 0,62, but with a SD from 0,04 to 1,28). It´s a very wide amplitud.”

  • We thank the reviewer for pointing out the wide confidence interval observed in the Cohen’s kappa for the variable SN.Puncture. As correctly noted, the 95% CI ranged from –0.045 to 1.276, despite a point estimate of 0.615. This wide range can be explained by two main factors: (1) the limited number of procedures included in this subsample (n = 10), and (2) the asymmetric distribution of the data, as only a few cases presented a positive outcome (sciatic nerve puncture = “yes”) in both ultrasound and dissection evaluations. These conditions reduce the statistical stability of the kappa coefficient and inflate the standard error, which directly affects the width of the confidence interval. This limitation has now been acknowledged in the discussion section of the manuscript.
    • 5. Limitations and Future Directions: Several limitations must be acknowledged in this study. First, the preservation state of the cadaveric specimens likely influenced the results. Compared to in vivo conditions, increased tissue stiffness and dehydration may have altered the mechanical resistance of deep soft tissues, potentially affecting needle trajectory and redirection during insertion. Second, in some specimens, particularly those stored in a prone position, muscle mass displacement such as flattening of the gluteus maximus made palpation of key anatomical landmarks (e.g., the lateral edge of the ischial tuberosity and the posterior margin of the greater trochanter) more challenging. These alterations may have led to minor deviations in the intended entry point, thereby compromising procedural accuracy. Additionally, anatomical variability among specimens, including differences in age, sex, BMI, and degree of muscular atrophy, may have influenced clinical decisions regarding needle size selection and perceived depth, as well as the ease of identifying anatomical landmarks and achieving accurate placement. Third, the characteristics of the needles used (0.30 mm in diameter, 30–60 mm in length) could have affected their behavior during insertion. In the presence of increased tissue resistance, longer needles are more prone to deflection, especially when targeting deep anatomical planes. Finally, another relevant aspect concerns the limited preparation time available to the physiotherapists before the procedure. The interventionists received the DDN protocol only 15 minutes prior to the intervention, without any opportunity for prior familiarization or rehearsal. Although both clinicians had experience with invasive procedures, the absence of prior training with the specific step-by-step protocol may have influenced their performance. Additionally, intra-rater reliability assessments were only performed on a subset of cadavers, as repeated insertions were limited to specimens in better preservation condition. This selective repetition, driven by both anatomical viability and operational constraints, may have introduced a sampling bias, as intra-rater reproducibility could not be evaluated across the full set of specimens Moreover, the interpretation of agreement coefficients should be approached with caution due to the small sample size (n = 5), which may have led to inflated standard errors and wide confidence intervals across all variables (for example, the CI observed in SN.Punct.), thereby reducing the statistical robustness of the reliability estimates. However, bilateral procedures were performed on all five cadavers, resulting in ten complete interventions, which represents a relatively robust design compared to previous cadaveric studies that often included fewer specimens, as highlighted in recent systematic reviews [24]. Future studies should ensure systematic repetition across all specimens and evaluate the impact of protocol exposure time on procedural outcomes, particularly when comparing training effects or skill acquisition under different conditions.”

In response to “The main thind a ask the authors to be done is to reference the discusion section. It´s imperative in order to get a scientific validity.”

  • Thank you for your valuable feedback. We have incorporated references into the discussion section and adapted some of the discussion points to improve the quality of the manuscript.

In response to “The main question addressed is whether a standardized, non-ultrasound-guided deep dry needling (DDN) protocol can safely and accurately target the quadratus femoris (QF) muscle in the ischiofemoral space (IFS), and whether the reliability of key procedural outcomes can be established between raters, using ultrasound and anatomical dissection as validation methods. The topic is relatively original and little relevant. It addresses a gap in the field: the safety and accuracy of DDN in the anatomically challenging ischiofemoral space without ultrasound guidance has not been well characterized. Given the clinical use of DDN in musculoskeletal pain management, providing empirical data on procedural safety, targeting accuracy, and reliability fills an important methodological and educational gap for physiotherapists. This study adds novel empirical data on the feasibility of a standardized, non-ultrasound-guided DDN approach to the QF muscle. It also offers methodological rigor by validating needle placement using both ultrasound imaging and anatomical dissection, and by quantifying intra- and inter-rater reliability. Such data are scarce in the current literature and can inform both training and clinical decision-making.”

  • We sincerely thank the reviewer for their thoughtful and encouraging feedback. We appreciate your recognition of the methodological value and clinical relevance of this study, particularly in addressing the safety and accuracy of deep dry needling to the quadratus femoris muscle without ultrasound guidance. As you correctly noted, the integration of ultrasound and dissection for needle placement validation, along with intra- and inter-rater reliability analysis, provides a structured and rigorous framework that is still underrepresented in the literature. We are pleased to know that these contributions were perceived as a valuable step toward bridging the methodological and educational gap in invasive physiotherapy techniques.

In response to “The sample size is very limited (n = 5 cadavers, 24 approaches), which may compromise generalizability; the authors should discuss this limitation more explicitly; Inter-rater reliability for needle size was poor (κ = 0.04); it would help to explain why and whether training can improve this; More detail on the criteria for “success” in needling, and on how the standardized protocol was taught to the raters, would enhance reproducibility; It may be useful to clarify how “IFS targeting” was operationally defined and measured; The selection of cadavers (e.g., age, sex, BMI) could be described to evaluate anatomical variability.”

  • Thank…
    • “2.3. DDN Procedure and Standardization Protocol: All needling procedures were performed by two licensed physiotherapists with differing levels of clinical experience in DDN and certified postgraduate training in invasive physiotherapy techniques. Experience levels were defined as intermediate (Ph.1, 5 years) and high (Ph.2, over 10 years). Both physiotherapists also held extensive teaching experience in human anatomy, serving as faculty in the Department of Physiotherapy at Universidad Europea de Madrid. Fifteen minutes before the intervention, physiotherapists received a written copy of the needling protocol to ensure consistency, which included detailed step-by-step instructions on needle placement. The document provided guidance on anatomical landmarks, measurement techniques for determining the entry point, and recommended needle inclination angles relative to cadaver positioning. Additional support was provided through visual aids, including multi-planar anatomical illustrations of the target region and diagrams of surrounding critical structures to avoid during insertion. To standardize protocol acquisition, a member of the research team presented the protocol aloud, clarified any questions raised by the clinicians, and highlighted key procedural steps. Visual support included on-screen images of the protocol displayed in the dissection room, allowing real-time review of anatomical references and technical instructions. Physiotherapists were also encouraged to ask questions during the needling procedures if clarification was required (Supplementary Material 1). Only two aspects were left to the clinician’s discretion: (1) the selection of the needle size and thickness based on cadaveric dimensions, and (2) the depth of needle insertion, which was evaluated exclusively for Ph. 2.”
    • “2.4. Ultrasonographic Assessment of DDN Procedure: Following each insertion, a physiotherapist with over 10 years of clinical experience in musculoskeletal ultrasound, who was blinded to physiotherapists needle insertion, performed a real-time sonographic evaluation using a high-resolution ultrasound system (Logiq S7 Expert, GE Healthcare, Chicago, IL, USA) to assess the success of DDN procedure. A single broad-spectrum linear matrix array probe (ML6-15, 50 mm field of view) with a frequency range of 5–15 MHz was used for all measurements. The following outcome variables were assessed: (1) needle tip position within the ischiofemoral space (success/failure); (2) needle tip position within the quadratus femoris muscle (success/failure); (3) vascular bundle puncture (success/failure); (4) sciatic nerve puncture (yes/no); and (5) needle size selected (0.30 × 30 mm; 0.30 × 40 mm; 0.30 × 50 mm; 0.30 × 60 mm). The first three were classified as dichotomous variables (success or failure), the fourth as a binary variable (yes/no), and the fifth as a multicategorical qualitative variable. The first three were classified as dichotomous variables (success or failure), the fourth as a binary variable (yes/no), and the fifth as a multicategorical qualitative variable. Needle placement within the ischiofemoral space (IFS) was operationally defined as the location of the needle tip between the lateral margin of the ischial tuberosity and the posterior edge of the greater trochanter, corresponding to the target region specified in the protocol. Insertions located outside this anatomical corridor, such as overlying bony structures or positioned proximally or distally to the defined boundaries, were considered failures. For the purposes of this study, global procedural success (Success = yes) was defined as those cases in which the needle tip reached the QF muscle, was located within IFS (specifically in the lateral region targeted by the needling approach, as detailed in the standardized protocol; Supplementary Material 1), and did not puncture the SN. Any deviation from these three criteria resulted in the classification of the procedure as unsuccessful. This operational definition was intended to reflect both anatomical accuracy and procedural safety. Quantitative ultrasound measurements were obtained using the system’s integrated electronic calipers and were expressed in centimeters, including: (6) distance from the needle tip to the sciatic nerve; (7) gluteal skin thickness; and (8) total depth of needle insertion (Figure 2). After completion of the sonographic evaluations, cadavers were preserved for subsequent anatomical dissection to confirm final needle placement. For the assessment of intra-rater reliability, Ph.2 repeated the procedures on cadavers 1 and 3, which corresponded to female and male specimens, respectively. This selection aimed to ensure sex parity and optimize specimen quality for evaluating procedural consistency. The final four interventions performed by Ph.2 on these cadavers were used for anatomical validation, with the needles left in situ to facilitate direct verification during dissection. To determine needle position, several ultrasound functionalities were employed, including the needle vision system, the virtual convex option to widen the insonation field and enhance lateral echo visualization, and adjustments to the insonation angle to optimize needle visibility. Additionally, power Doppler was activated while vibrating the needle to amplify signal detection and accurately localize its position (Supplementary Material 2).”
    • “4.2. Safety, Accuracy and Reliability of the Procedure For the multicategorical variable of needle size selection, intra-rater reproducibility was limited (50% agreement; mode frequency = 50%), as Ph.2 selected a different needle in the second round. This adjustment may reflect clinical reasoning based on perceived tissue depth or insertion resistance. In contrast, inter-rater agreement was 0%, with Ph.1 consistently selecting the intermediate lenght (0.30×40 mm), while Ph.2 opted for the longest needle (0.30×60 mm). These differences suggest that material selection may be influenced by each clinician’s spatial judgment and professional experience, as well as subjective risk perception. Beyond the reliability analysis, a noteworthy observation was identified during the review of individual cases, as Ph.2 selected a shorter 0.30×30 mm needle in Cadaver 4 despite the anatomical depth of the target muscle. This isolated choice may be explained by the condition of the specimen (height: 1.62 m; weight: 55 kg), which was notably the lowest among all cadavers. Due to severe muscular atrophy and pronounced tissue flattening, bony landmarks were more prominent and superficially palpable than in typical in vivo conditions. This altered perception could have led the clinician to underestimate the required depth and opt for a shorter needle. Such considerations highlight the methodological limitation of extrapolating needling choices in preserved specimens to real-world clinical practice, especially when dealing with older cadavers exhibiting substantial anatomical degradation. Importantly, the lack of prior consensus or training on needle size selection was intentionally incorporated into the study design to explore individual decision-making under realistic clinical conditions. This approach likely contributed to the poor inter-rater reliability observed (κ = 0.04), underscoring the need for standardized selection criteria in future applications. With regard to the dichotomous outcomes, the agreement observed was high (80%) for both IFS location and SN.Punct., although κ was 0.000 in both cases due to the absence of variability in one rater’s assessments. Notably, the 80% agreement for SN.Punct. was entirely attributable to instances where both evaluators reported no SN puncture, thereby supporting the overall safety of the approach. Additionally, none of the reliability comparisons yielded statistically significant differences (all p ≥ 0.301), further confirming the consistency of the observed patterns across raters.”
    • 5. Limitations and Future Directions: Several limitations must be acknowledged in this study. First, the preservation state of the cadaveric specimens likely influenced the results. Compared to in vivo conditions, increased tissue stiffness and dehydration may have altered the mechanical resistance of deep soft tissues, potentially affecting needle trajectory and redirection during insertion. Second, in some specimens, particularly those stored in a prone position, muscle mass displacement such as flattening of the gluteus maximus made palpation of key anatomical landmarks (e.g., the lateral edge of the ischial tuberosity and the posterior margin of the greater trochanter) more challenging. These alterations may have led to minor deviations in the intended entry point, thereby compromising procedural accuracy. Additionally, anatomical variability among specimens, including differences in age, sex, BMI, and degree of muscular atrophy, may have influenced clinical decisions regarding needle size selection and perceived depth, as well as the ease of identifying anatomical landmarks and achieving accurate placement. Third, the characteristics of the needles used (0.30 mm in diameter, 30–60 mm in length) could have affected their behavior during insertion. In the presence of increased tissue resistance, longer needles are more prone to deflection, especially when targeting deep anatomical planes. Finally, another relevant aspect concerns the limited preparation time available to the physiotherapists before the procedure. The interventionists received the DDN protocol only 15 minutes prior to the intervention, without any opportunity for prior familiarization or rehearsal. Although both clinicians had experience with invasive procedures, the absence of prior training with the specific step-by-step protocol may have influenced their performance. Additionally, intra-rater reliability assessments were only performed on a subset of cadavers, as repeated insertions were limited to specimens in better preservation condition. This selective repetition, driven by both anatomical viability and operational constraints, may have introduced a sampling bias, as intra-rater reproducibility could not be evaluated across the full set of specimens Moreover, the interpretation of agreement coefficients should be approached with caution due to the small sample size (n = 5), which may have led to inflated standard errors and wide confidence intervals across all variables (for example, the CI observed in SN.Punct.), thereby reducing the statistical robustness of the reliability estimates. However, bilateral procedures were performed on all five cadavers, resulting in ten complete interventions, which represents a relatively robust design compared to previous cadaveric studies that often included fewer specimens, as highlighted in recent systematic reviews [24]. Future studies should ensure systematic repetition across all specimens and evaluate the impact of protocol exposure time on procedural outcomes, particularly when comparing training effects or skill acquisition under different conditions.”

In response to “The conclusions are generally consistent with the evidence. The authors appropriately qualify their findings by describing accuracy as moderate and safety as acceptable, while acknowledging limited inter-rater reliability. They also correctly highlight the high agreement between ultrasound and dissection, supporting their combined use in validation. However, it would strengthen the discussion to more explicitly address the clinical implications of SN puncture risk and limited reliability.”

  • Thank you for your thoughtful comment. We have revised the Discussion and Practical Applications sections to more clearly address the clinical implications of the observed SN puncture rate and the limited inter-rater reliability. Specifically, we now emphasize the potential risks associated with blind application of this technique, highlight the need for advanced anatomical knowledge and structured training, and propose strategies to enhance procedural consistency. These points are also reflected in the adapted Conclusions section, which now better conveys the moderate anatomical accuracy and the safety limitations of the protocol.
    • Conclusions: A standardized, non-ultrasound-guided DDN protocol to the QF muscle demonstrated moderate anatomical accuracy and limited safety in cadaveric specimens. Despite the use of a step-by-step approach, the 29.2% of procedures failed to reach the IFS and 16.7% resulted in sciatic nerve puncture, indicating a relevant clinical risk. Intra- and inter-rater reliability was also limited, particularly for needle size selection, reflecting the complexity of consistent decision-making in blind procedures. The high agreement between ultrasound and dissection underscores the value of imaging validation for training, while detailed anatomical protocols remain essential when imaging is unavailable.”
    • 6. Practical Applications: The findings of this study provide valuable insights for clinical and educational practice regarding the implementation of DDN in anatomically complex regions such as the deep gluteal space. While the procedure showed partial anatomical success and a moderate rate of sciatic nerve avoidance, nearly one-third of the procedures failed to reach the IFS, and nerve puncture occurred in almost one out of every six attempts. These results highlight the need for extreme caution in blind applications of this technique. In settings where imaging is not available, the implementation of detailed, step-by-step protocols based on reliable anatomical landmarks, optimized needle direction, angle, and depth, may help reduce variability and enhance procedural safety. These protocols may serve as a valuable educational tool, particularly for clinicians with limited experience in blind needling approaches. Furthermore, the integration of dual validation through ultrasound and dissection, along with intra- and inter-rater reliability analysis, offers a methodological model for assessing procedural consistency and refining educational protocols. Notably, the inclusion of an external observer to ensure adherence to protocol strengthens the quality control of the intervention. This study illustrates how structured methodological designs can contribute to the development of safer and more accurate standardized approaches for invasive physiotherapy techniques, especially when applied to anatomically sensitive regions such as the deep gluteal space. Given the 16.7% rate of sciatic nerve puncture observed, clinical application of this approach may require advanced anatomical knowledge and should ideally be supported by imaging when available. Additionally, the low inter-rater reliability in needle selection underscores the potential for variability in clinical decision-making, emphasizing the importance of enhanced training strategies, the use of anatomical reference tools, and structured protocols to promote consistent and safe implementation in clinical settings. These findings are not intended to promote the routine clinical use of unguided deep dry needling, but to inform future research and training in anatomically complex regions.”

In response to “While the abstract does not list specific references, the topic demands up-to-date references on DDN safety, ischiofemoral anatomy, and validation methods using imaging and dissection. The authors should ensure these are included and reflect recent systematic reviews or guidelines where available.”

  • We appreciate the reviewer’s feedback and agree that a more thorough contextualization was warranted. Accordingly, we have enriched the Discussion section by incorporating additional references from the current literature to support key arguments and strengthen the interpretative framework. These references include recent studies addressing the reliability of ultrasound in cadaveric models, challenges associated with needle visualization in out-of-plane approaches, and the clinical implications of non-standardized dry needling protocols. Furthermore, we have expanded several discussion points to offer a more comprehensive interpretation of our findings, particularly regarding safety concerns, technical limitations, and the relevance of procedural standardization. These additions aim to enhance the clarity, depth, and clinical relevance of the manuscript. All changes are now reflected in section 4, with new citations clearly indicated.

Sánchez-Montoya M, Almazán-Polo J, González-de-la-Flor Á. Safety and anatomical accuracy of dry needling in musculoskeletal system: a systematic review of cadaveric studies. J Man Manip Ther. Forthcoming 2025. doi:10.1080/10669817.2025.2536818.

Valera-Calero JA, Varol U, Plaza-Manzano G, Fernández-de-las-Peñas C, Belón-Pérez P, López-Redondo M, et al. Testing the Safety of Piriformis Dry Needling Interventions: An Observational Study Evaluating the Predictive Value of Anthropometric and Demographic Factors. Journal of Clinical Medicine 2024, Vol 13, Page 6674 2024;13:6674. https://doi.org/10.3390/JCM13226674.

Sánchez-Montoya M, Almazán-Polo J, González-de-la-Flor Á. Safety and Anatomical Accuracy of Dry Needling in Musculoskeletal System: A Systematic Review of Cadaveric Studies. Journal of Manual & Manipulative Therapy 2025.

Valera-Calero JA, Varol U, Plaza-Manzano G, Fernández-de-las-Peñas C, Belón-Pérez P, López-Redondo M, et al. Testing the Safety of Piriformis Dry Needling Interventions: An Observational Study Evaluating the Predictive Value of Anthropometric and Demographic Factors. Journal of Clinical Medicine 2024, Vol 13, Page 6674 2024;13:6674. https://doi.org/10.3390/JCM13226674.

Duijn EAHD, Roy van S, Karel YHJM, Provyn S, Pouliart N. An interexaminer agreement and reliability study on cadavers with musculoskeletal ultrasound of the shoulder performed by physiotherapists and radiologists compared with dissection. Musculoskelet Sci Pract 2022;60. https://doi.org/10.1016/j.msksp.2022.102569.

Vitt M, Macaraeg S, Stapleton Z, Mata A, Ross BS. Ultrasound verification of palpation-based dry needling techniques of rotator cuff muscles: a prospective feasibility trial. Journal of Manual and Manipulative Therapy 2024;32:166–72. https://doi.org/10.1080/10669817.2023.2244735,.

Dinges HC, Hoeft J, Cornelius VM, Steinfeldt T, Wiesmann T, Wulf H, et al. Nominal logistic regression analysis of variables determining needle visibility in ultrasound images – a full factorial cadaver study. BMC Anesthesiol 2023;23:1–8. https://doi.org/10.1186/S12871-023-02339-Y/TABLES/3.

Kearns GA, Brismée JM, Riley SP, Wang-Price S, Denninger T, Vugrin M. Lack of standardization in dry needling dosage and adverse event documentation limits outcome and safety reports: a scoping review of randomized clinical trials. Journal of Manual and Manipulative Therapy 2023;31:72–83. https://doi.org/10.1080/10669817.2022.2077516,.

In response to “The abstract does not provide details on tables or figures. In the full manuscript, the authors should ensure that tables summarizing reliability metrics (e.g., Cohen’s kappa, Valence Index) are clear and include confidence intervals if possible.”

  • Thank you for your feedback. We have removed the reference to the reliability indices mentioned in the Abstract in accordance with your opinion. We also modified Table 2 by splitting it into Tables 2a and 2b to improve the interpretation of the results. Additionally, we incorporated p-values and the number of observations into the tables. Confidence intervals were reported for all analyses in which it was possible to evaluate or interpret Cohen’s kappa.
    • “Introduction: Deep dry needling (DDN) is commonly applied in physiotherapy to treat musculoskeletal pain. The quadratus femoris (QF) muscle, located in the ischiofemoral space (IFS), represents a clinically relevant yet anatomically complex target. However, limited evidence exists on the safety, accuracy, and reliability of non-ultrasound-guided DDN in this region. Aims: To assess the safety and accuracy of a standardized, non-ultrasound-guided DDN approach to the QF muscle, and to evaluate intra- and inter-rater reliability of key procedural outcomes. Additionally, to determine the agreement between ultrasound imaging and anatomical dissection as validation methods for needle placement. Methods: An experimental cross-sectional study was conducted on five fresh cadavers (n = 24 approaches) by two physiotherapists with different DN experience. A standardized dry needling protocol was executed without ultrasound guidance, and anatomical and procedural variables were documented. Reliability (intra/inter-rater) was assessed for needle size, sciatic nerve (SN) puncture, IFS targeting, and overall success. In a subset, needle placement was validated through ultrasound and subsequent dissection. Results: The IFS was reached in 70.8% of procedures, and the SN was punctured in 16.7%. Inter-rater reliability for needle size was poor (κ = 0.04). Agreement between ultrasound and dissection was excellent for ischiofemoral location and success (100%) and moderate for non SN puncture (90%; κ = 0.62). Conclusion: The standardized protocol demonstrated moderate accuracy and revealed a relevant clinical risk when targeting the quadratus femoris muscle. While inter-rater reliability was limited, agreement between ultrasound and dissection methods was high, supporting their complementary use for validating needle placement in anatomically complex procedures.”

In response to “Any images of ultrasound or dissection validation should be well-labeled and high-resolution to support interpretation.”

  • We have attached the image in PNG format to improve its resolution and quality according to your suggestion. We appreciate the opportunity to improve the quality of the manuscript.

Thanks for your valuable commentaries which have permitted us to improve the quality of the manuscript.

Sincerely,

The authors

Reviewer 3 Report

Comments and Suggestions for Authors

Dear researchers,

I address my comments and suggestions regarding your valuable research:

The study addresses a highly relevant topic, particularly in the context of innovative therapeutic approaches such as dry needling, especially when targeting deep muscle layers. The agreement between ultrasound imaging and anatomical dissection is essential for validating adherence to safety principles, correct needle placement, and the effectiveness of therapeutic interventions.

One of the study's striking findings is the 29.2% failure rate, notably among physiotherapists with 5–10 years of experience. Given their likely familiarity with both superficial and deep muscular dry needling, this result suggests that treating live patients—particularly those experiencing pain—introduces additional layers of complexity and risk. The anatomical region exposed, along with necessary precautions, must be carefully considered when planning such interventions.

The study confirms important methodological and practical aspects of performing dry needling. However, the generalizability of these findings is uncertain. With only two therapists included, it's unclear whether the conclusions can be confidently extended to broader clinical practice.

Moreover, the validation of needle placement on cadavers, while valuable, offers only partial confirmation. Limitations in intra- and inter-rater reliability reduce the certainty of the conclusions. This issue, coupled with the use of only two comparative studies, raises questions about the sufficiency of evidence to support the routine, ultrasound-free use of deep dry needling.

A deeper literature review and comparison with other studies—particularly those that emphasize how many experienced therapists rely on ultrasound for such procedures—would strengthen the article’s conclusions. More comprehensive evidence is needed to assess whether ultrasound guidance is essential for both safety and effectiveness in deep dry needling.

In summary, while the study provides valuable insights and confirms certain practical aspects of dry needling, it also highlights areas of uncertainty and risk, especially in real-life patient contexts. Further research with a larger pool of practitioners and broader methodology is warranted before fully endorsing independent, unguided application of deep dry needling at scale.

Best regards.

Author Response

Responses to Reviewers’s comments:

Response to Reviewer 3 comment:

In response to “Dear researchers, I address my comments and suggestions regarding your valuable research:The study addresses a highly relevant topic, particularly in the context of innovative therapeutic approaches such as dry needling, especially when targeting deep muscle layers. The agreement between ultrasound imaging and anatomical dissection is essential for validating adherence to safety principles, correct needle placement, and the effectiveness of therapeutic interventions.”

  • We sincerely thank the reviewer for their positive and encouraging feedback. We fully agree that validating dry needling procedures through both ultrasound imaging and anatomical dissection is essential to ensure safety, accuracy, and clinical effectiveness. We appreciate the recognition of our work's relevance within this context.

In response to “One of the study's striking findings is the 29.2% failure rate, notably among physiotherapists with 5–10 years of experience. Given their likely familiarity with both superficial and deep muscular dry needling, this result suggests that treating live patients—particularly those experiencing pain—introduces additional layers of complexity and risk. The anatomical region exposed, along with necessary precautions, must be carefully considered when planning such interventions.”

  • We appreciate the reviewer’s thoughtful observation regarding the reported 29.2% failure rate and its implications. We agree that this finding highlights the potential challenges of performing deep dry needling even among clinicians with relevant experience, and we acknowledge the increased complexity when extrapolating these procedures to live patients. In response, we have taken this valuable consideration into account and revised the discussion to further emphasize the influence of anatomical complexity, patient-related factors, and clinical decision-making in high-risk regions. The text has been modified accordingly to reflect this appreciation.
    • 1. Overview of finfings: This study aimed to evaluate the accuracy and safety of a DDN approach to the QF muscle through a structured and standardized step-by-step protocol in a cadaveric model. The IFS was successfully reached in 70.8% of procedures, while SN puncture occurred in 16.7% of cases. Although these findings reflect a potentially acceptable anatomical profile in terms of safety, the puncture rate may still be considered clinically relevant. Moreover, the fact that nearly one-third of insertions failed to reach the intended target highlights important limitations in procedural accuracy, even under controlled conditions. These results underscore the need for cautious interpretation when considering the implementation of blind techniques in clinical practice. To the best of the authors’ knowledge, this is the first study to systematically evaluate the reliability of DDN procedures targeting the QF muscle in a cadaveric model, while simultaneously validating procedural outcomes through dual assessment using both ultrasound imaging and anatomical dissection. Although previous studies have employed both modalities to verify needle positioning [11,22], none have specifically aimed to assess the inter-method agreement between ultrasound and dissection alongside inter-examiner reliability metrics. By combining reproducibility metrics across evaluators with differing levels of clinical experience and inter-method agreement between imaging and dissection, the present findings provide novel insights into the technical feasibility, anatomical risks, and standardization challenges associated with invasive procedures in high-risk anatomical zones. The absence of prior data for direct comparison underscores the originality of this approach and highlights the need for further research into adjunct strategies such as ultrasound guidance, protocol refinement, and targeted training programs to improve procedural accuracy, safety, and reproducibility in both clinical and educational contexts. In recent years, several cadaveric studies have described invasive approaches to anatomically challenging regions, primarily focusing on needle placement accuracy in relation to target tissues and critical neurovascular structures. These investigations have included muscles such as the pronator quadratus, pronator teres, supinator, lateral pterygoid, popliteus, piriformis, and tibialis posterior, among others [6,10,13,15,22,23]. While this methodological approach constitutes a strength of the present study, offering a novel contribution to the literature, it simultaneously limits the possibility of direct comparisons with prior research. Nevertheless, these findings align with recent evidence supporting the safety of non-guided DDN when standardized protocols are followed, although ultrasound may be advisable in high-risk regions due to methodological limitations in previous studies [24].”
    • Conclusions: A standardized, non-ultrasound-guided DDN protocol to the QF muscle demonstrated moderate anatomical accuracy and limited safety in cadaveric specimens. Despite the use of a step-by-step approach, the 29.2% of procedures failed to reach the IFS and 16.7% resulted in sciatic nerve puncture, indicating a relevant clinical risk. Intra- and inter-rater reliability was also limited, particularly for needle size selection, reflecting the complexity of consistent decision-making in blind procedures. The high agreement between ultrasound and dissection underscores the value of imaging validation for training, while detailed anatomical protocols remain essential when imaging is unavailable.”
    •  

In response to “The study confirms important methodological and practical aspects of performing dry needling. However, the generalizability of these findings is uncertain. With only two therapists included, it's unclear whether the conclusions can be confidently extended to broader clinical practice.”

  • We thank the reviewer for this thoughtful observation. We agree that including only two therapists limits the generalizability of the findings. In response, we have explicitly acknowledged this limitation in the manuscript and clarified that the results should be interpreted with caution when extrapolating to broader clinical practice.
    • “4.3 Limitations and Future Directions: The interventionists received the DDN protocol only 15 minutes prior to the intervention, without any opportunity for prior familiarization or rehearsal. Although both clinicians had experience with invasive procedures, the absence of prior training with the specific step-by-step protocol may have influenced their performance. The inclusion of only two physiotherapists limits the generalizability of the findings, which should be interpreted with caution when extrapolating to broader clinical practice.

In response to “Moreover, the validation of needle placement on cadavers, while valuable, offers only partial confirmation. Limitations in intra- and inter-rater reliability reduce the certainty of the conclusions. This issue, coupled with the use of only two comparative studies, raises questions about the sufficiency of evidence to support the routine, ultrasound-free use of deep dry needling.”

  • We thank the reviewer for this insightful comment. We fully acknowledge that cadaveric validation provides only a partial representation of clinical conditions, and that the absence of dynamic tissue properties and patient-related factors limits its external validity. Our aim was not to recommend the routine use of ultrasound-free deep dry needling, but rather to explore its anatomical feasibility and safety under controlled conditions using a standardized protocol. Regarding reliability, we agree that the limited intra- and inter-rater agreement—particularly in needle size selection—highlights the need for enhanced training and clearer procedural criteria. Finally, while only a few previous studies have used dual validation methods (ultrasound and dissection), our work adds to this limited body of literature by incorporating both reliability metrics and inter-method agreement. We have clarified these points in the revised Discussion and Conclusion sections to avoid overinterpretation of the findings.

In response to “A deeper literature review and comparison with other studies—particularly those that emphasize how many experienced therapists rely on ultrasound for such procedures—would strengthen the article’s conclusions. More comprehensive evidence is needed to assess whether ultrasound guidance is essential for both safety and effectiveness in deep dry needling.”

  • We appreciate the reviewer’s feedback and agree that a more thorough contextualization was warranted. Accordingly, we have enriched the Discussion section by incorporating additional references from the current literature to support key arguments and strengthen the interpretative framework. These references include recent studies addressing the reliability of ultrasound in cadaveric models, challenges associated with needle visualization in out-of-plane approaches, and the clinical implications of non-standardized dry needling protocols. Furthermore, we have expanded several discussion points to offer a more comprehensive interpretation of our findings, particularly regarding safety concerns, technical limitations, and the relevance of procedural standardization. These additions aim to enhance the clarity, depth, and clinical relevance of the manuscript. All changes are now reflected in section 4, with new citations clearly indicated.

Sánchez-Montoya M, Almazán-Polo J, González-de-la-Flor Á. Safety and anatomical accuracy of dry needling in musculoskeletal system: a systematic review of cadaveric studies. J Man Manip Ther. Forthcoming 2025. doi:10.1080/10669817.2025.2536818.

Valera-Calero JA, Varol U, Plaza-Manzano G, Fernández-de-las-Peñas C, Belón-Pérez P, López-Redondo M, et al. Testing the Safety of Piriformis Dry Needling Interventions: An Observational Study Evaluating the Predictive Value of Anthropometric and Demographic Factors. Journal of Clinical Medicine 2024, Vol 13, Page 6674 2024;13:6674. https://doi.org/10.3390/JCM13226674.

Sánchez-Montoya M, Almazán-Polo J, González-de-la-Flor Á. Safety and Anatomical Accuracy of Dry Needling in Musculoskeletal System: A Systematic Review of Cadaveric Studies. Journal of Manual & Manipulative Therapy 2025.

Valera-Calero JA, Varol U, Plaza-Manzano G, Fernández-de-las-Peñas C, Belón-Pérez P, López-Redondo M, et al. Testing the Safety of Piriformis Dry Needling Interventions: An Observational Study Evaluating the Predictive Value of Anthropometric and Demographic Factors. Journal of Clinical Medicine 2024, Vol 13, Page 6674 2024;13:6674. https://doi.org/10.3390/JCM13226674.

Duijn EAHD, Roy van S, Karel YHJM, Provyn S, Pouliart N. An interexaminer agreement and reliability study on cadavers with musculoskeletal ultrasound of the shoulder performed by physiotherapists and radiologists compared with dissection. Musculoskelet Sci Pract 2022;60. https://doi.org/10.1016/j.msksp.2022.102569.

Vitt M, Macaraeg S, Stapleton Z, Mata A, Ross BS. Ultrasound verification of palpation-based dry needling techniques of rotator cuff muscles: a prospective feasibility trial. Journal of Manual and Manipulative Therapy 2024;32:166–72. https://doi.org/10.1080/10669817.2023.2244735,.

Dinges HC, Hoeft J, Cornelius VM, Steinfeldt T, Wiesmann T, Wulf H, et al. Nominal logistic regression analysis of variables determining needle visibility in ultrasound images – a full factorial cadaver study. BMC Anesthesiol 2023;23:1–8. https://doi.org/10.1186/S12871-023-02339-Y/TABLES/3.

Kearns GA, Brismée JM, Riley SP, Wang-Price S, Denninger T, Vugrin M. Lack of standardization in dry needling dosage and adverse event documentation limits outcome and safety reports: a scoping review of randomized clinical trials. Journal of Manual and Manipulative Therapy 2023;31:72–83. https://doi.org/10.1080/10669817.2022.2077516,.

In response to “In summary, while the study provides valuable insights and confirms certain practical aspects of dry needling, it also highlights areas of uncertainty and risk, especially in real-life patient contexts. Further research with a larger pool of practitioners and broader methodology is warranted before fully endorsing independent, unguided application of deep dry needling at scale.”

  • We appreciate the reviewer’s summary and fully agree that, although the study offers valuable anatomical and methodological insights, its findings should be interpreted with caution in clinical contexts. In response, we have clarified in the “Practical Applications” section that the results are not intended to support the routine clinical use of unguided deep dry needling, but rather to inform future research and training efforts.
    • 6. Practical Applications: Given the 16.7% rate of sciatic nerve puncture observed, clinical application of this approach may require advanced anatomical knowledge and should ideally be supported by imaging when available. Additionally, the low inter-rater reliability in needle selection underscores the potential for variability in clinical decision-making, emphasizing the importance of enhanced training strategies, the use of anatomical reference tools, and structured protocols to promote consistent and safe implementation in clinical settings [24]. These findings are not intended to promote the routine clinical use of unguided deep dry needling, but to inform future research and training in anatomically complex regions.”

Thanks for your valuable commentaries which have permitted us to improve the quality of the manuscript.

Sincerely,

The authors
